# A less cloudy picture of the inter-model spread in future global warming projections

Xiaoming Hu [1,2], Hanjie Fan [1], Ming Cai [3✉], Sergio A. Sejas [4], Patrick Taylor [5] & Song Yang [1,2]

Model warming projections, forced by increasing greenhouse gases, have a large inter-model spread in both their geographical warming patterns and global mean values. The inter-model warming pattern spread (WPS) limits our ability to foresee the severity of regional impacts on nature and society. This paper focuses on uncovering the feedbacks responsible for the WPS. Here, we identify two dominant WPS modes whose global mean values also explain 98.7% of the global warming spread (GWS). We show that the ice-albedo feedback spread explains uncertainties in polar regions while the water vapor feedback spread explains uncertainties elsewhere. Other processes, including the cloud feedback, contribute less to the WPS as their spreads tend to cancel each other out in a model-dependent manner. Our findings suggest that the WPS and GWS could be significantly reduced by narrowing the inter-model spreads of ice-albedo and water vapor feedbacks, and better understanding the spatial coupling between feedbacks.

[1] School of Atmospheric Sciences, Sun Yat-sen University, 510275 Guangzhou, China. [2] Southern Marine Science and Engineering Guangdong Laboratory (Zhuhai), 519082 Zhuhai, China. [3] Department of Earth, Ocean and Atmospheric Sciences, Florida State University, Tallahassee, FL 32304, USA. [4] Science Systems and Applications, Inc, Hampton, VA 23666, USA. [5] Climate Science Branch, NASA Langley Research Center, Hampton, VA 23681, USA. ✉email: mcai@fsu.edu

The Intergovernmental Panel on Climate Change (IPCC) fifth assessment report (AR5) projects that under the RCP8.5 (Representative Concentration Pathway) warming scenario, the global-mean surface temperature will likely warm between 2.6 and 4 K by the end of the twenty-first century relative to the end of the twentieth century. The substantial range or spread of the global-mean warming projection reflects the uncertainty of the warming amount in future climate projections by climate models and limits our ability to foresee the severity of the global warming impacts on nature and human civilization[1–3]. A quantitative evaluation of the physical mechanisms responsible for the inter-model differences in the global-mean warming projection (referred to as global warming spread and abbreviated as GWS hereafter) is key to understanding why models exhibit varying degrees of warming when forced by the same increase of greenhouse gases. In the literature, the inter-model differences in cloud feedback strength have been argued as the main cause of the GWS[4–9]. Water vapor feedback uncertainty was also shown to be a primary contributor to the GWS, but when combined with the lapse-rate feedback, as is commonly done, the contribution diminished substantially[10].

The GWS is accompanied by the inter-model spread of geographic warming patterns, as the amount of warming varies greatly from location to location. Arctic surface warming, for example, is three to four times greater than the global-mean warming[2,11,12]. For an easy reference, we refer to the inter-model spread of geographic warming patterns as the warming pattern spread (WPS). The central question of this study is to explore whether the GWS can be explained by a few dominant patterns of the WPS. If so, the task of identifying the factors giving rise to the GWS becomes revealing the feedback processes that are responsible for the WPS. There are a few studies relating the spatial patterns of the inter-model WPS to changes in clouds, albedo, water vapor, and other factors[13,14]. Cloud feedback was shown to be the dominant contributor to the warming uncertainty in the tropics[10], followed by the contribution of the combined water vapor and lapse-rate feedbacks. The large Arctic warming spread is found to be mainly driven by local feedbacks, with the uncertainty in the surface albedo and lapse-rate feedbacks as the primary contributors[1,2,15].

Most of the aforementioned studies use a top-of-atmosphere (TOA) perspective to evaluate process contributions to the GWS[4–7,10]. Recently, other studies[1,16–22] have used a surface perspective to evaluate feedback contributions to the projected surface warming based on the surface energy balance equation. The surface perspective has the benefit of including the effects of non-radiative processes such as the ocean and atmospheric dynamics, and surface turbulent heat fluxes in addition to the radiative processes. The surface perspective has been shown to provide a valuable alternative interpretation of feedback contributions to surface warming[1,21]. Moreover, the majority of work in the literature focus on climate feedback parameters, which are defined as the TOA radiative flux changes associated with individual feedback processes normalized by global-mean temperature change, in searching for the main sources of the global-mean climate projection uncertainty using climate models.

Here, we directly compare the WPS and GWS with inter-model spreads of spatial patterns and global means of partial energy flux perturbations at the TOA associated with internal processes or their equivalents at the surface by converting partial energy flux perturbations at the surface to partial surface temperature changes, without any normalization. Our analysis reveals two dominant WPS patterns whose global means explain 98.7% of the GWS. One of the dominant WPS patterns indicates models with greater (smaller) global-mean warming tend to have larger (lesser) warming everywhere with a more (less) pronounced polar warming amplification signal, particularly over the Arctic. The other WPS pattern indicates some models with greater (smaller) global-mean warming tend to have greater (less) warming in the tropics with less (more) polar warming amplification. Our study further reveals that the two patterns are mainly driven by the inter-model spreads of the ice-albedo and water vapor feedbacks with the former being responsible for the uncertainties in polar regions and the latter responsible for uncertainties elsewhere. The cloud feedback spread, despite its large amplitude, contributes less to the GWS because it tends to be canceled out by different model-dependent combinations of other feedback spreads that are not well correlated with the dominant WPS patterns.

## Results

**Spreads of global warming projections and climate feedbacks.** The multi-model ensemble (MME) mean surface warming response under the RCP8.5 scenario displays the characteristic polar warming amplification (PWA) pattern (Fig. 1a). Though this characteristic pattern persists across the CMIP5 (Coupled Model Intercomparison Project Phase 5) model ensemble, both the global mean and regional warming amounts exhibit substantial differences across models, particularly in polar regions (Fig. 1b). We define the WPS as the departures of individual models' zonal mean temperature changes from the MME (Fig. 1c). The global mean of the WPS corresponds to the GWS. We here adopt the standard textbook definition of feedback, which is defined as a change of energy input/output resulting from an internal process that in turn either amplifies or opposes the initial perturbation in energy input/output caused by the external forcing. Using the climate feedback-response analysis method (CFRAM)[23,24], we evaluate individual process contributions to the spatial warming pattern for each of the CMIP5 models. Specifically, we calculate the (partial) surface temperature changes required to balance the surface radiative or non-radiative energy flux perturbations caused by the greenhouse gas forcing and feedbacks (Supplementary Fig. 1). Removing the MME mean for each of the partial surface temperature changes given by the forcing and feedbacks for every model, reveals the inter-model spread associated with individual processes (Fig. 1d–k). It is seen that the sum of the inter-model spreads of these partial temperature changes is approximately equal to the (total) inter-model WPS, except for the notable differences over the Southern Ocean and Arctic Ocean (Fig. 1c versus Fig. 1l or Fig. 1m). This equivalence allows us to quantify individual process contributions to the inter-model WPS.

The (partial) surface temperature changes due to ocean dynamics plus ocean heat storage (OCN), surface turbulent heat fluxes (HF), and clouds (CLD) display large spreads at all latitudes (Fig. 1d–k). The partial temperature change due to the ice-albedo feedback displays a large spread in polar regions. Assuming that climate feedbacks are independent of one another, the feedback with the largest global-mean spread would unequivocally be the primary contributor to GWS. Climate feedbacks, however, are dependent on each other (i.e., coupled). As a result, the inter-model uncertainty of one climate feedback is linked to the uncertainty of the others. Therefore, the feedback(s) with the larger spread(s) in their global means may not be the largest contributors to the (total) inter-model GWS. A correlation analysis between the global-mean spreads of individual feedbacks and the GWS shows that feedbacks with the largest spread, as measured by the standard deviation, do not necessarily correlate well with the GWS (Supplementary Fig. 2). For example, the temperature spread associated with changes in ocean dynamics plus heat storage (OCN), despite its large amplitude, weakly correlates with the GWS. On the other hand, the global-mean albedo spread is small yet strongly correlates with the GWS.

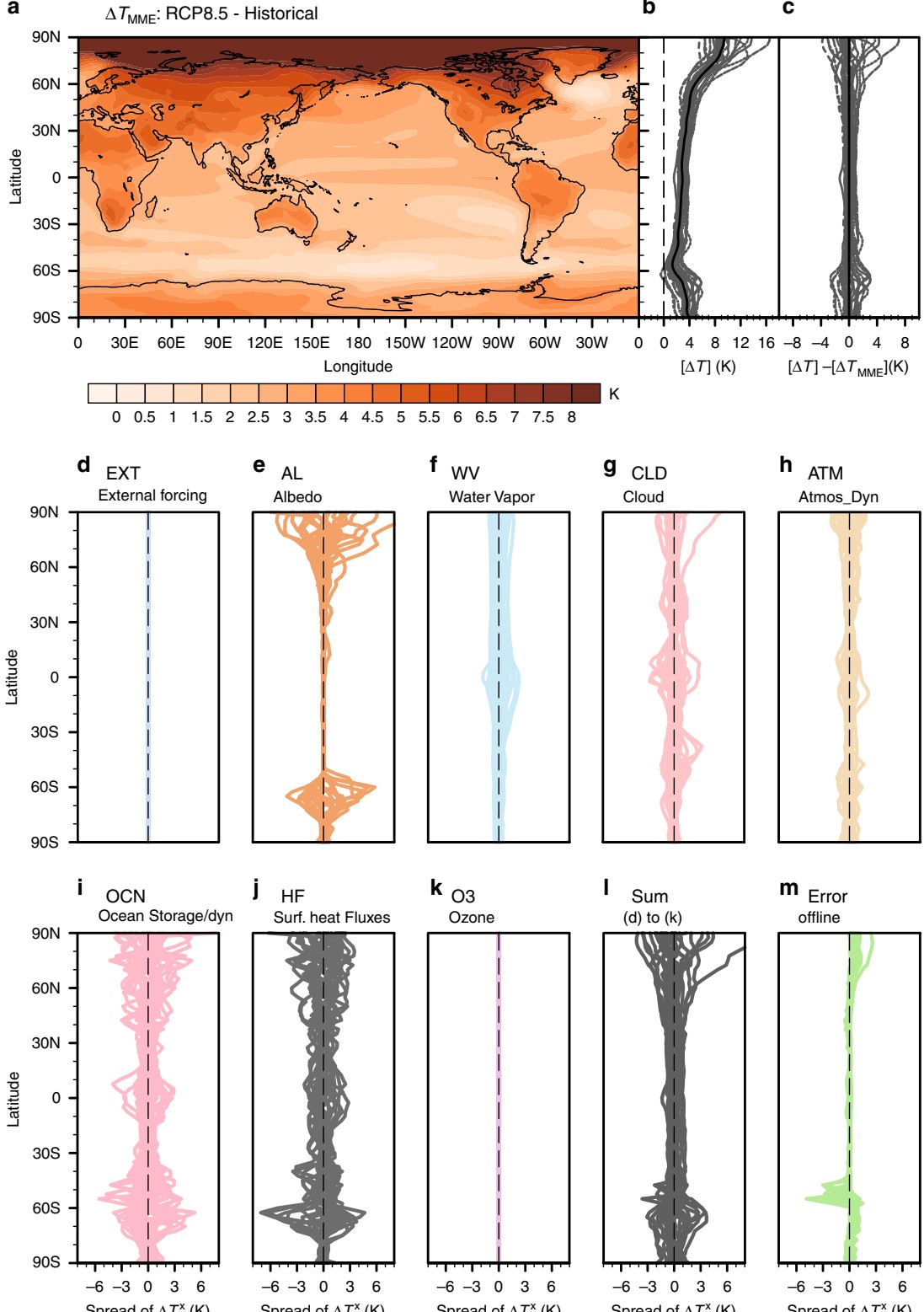

**Fig. 1 Inter-model spread of surface warming. a** Multi-model ensemble (MME) mean surface temperature changes ($K$) between the projected climate (2051–2100) of the RCP8.5 (Representative Concentration Pathway) simulations and the historical climate (1951–2000) derived from 25 models of the Coupled Model Intercomparison Project Version 5. **b** Zonal mean surface temperature change ($K$) for individual models (gray lines) and their MME mean (thick black line). **c** Same as **b** but with the MME removed. Inter-model surface warming spreads (with their respective MMEs removed) in the individual partial surface temperature changes ($K$) due to changes in **d** carbon dioxide (EXT), **e** surface albedo (AL), **f** water vapor (WV), **g** clouds (CLD), **h** atmospheric dynamics (ATM), **i** ocean dynamics plus heat storage (OCN), **j** surface turbulent heat fluxes (HF), and **k** ozone (O3). **l** The sum of **d**–**k** and **m** is the difference between **l** and **c**, corresponding to the error of the CFRAM analysis. The square bracket [] in the abscissa labels of **b** and **c** represents the zonal average of the surface warming whereas the superscript X in the abscissa labels of **d**–**m** denotes the zonal average of the partial temperature changes associated with processes whose names are given on the top of the panels.

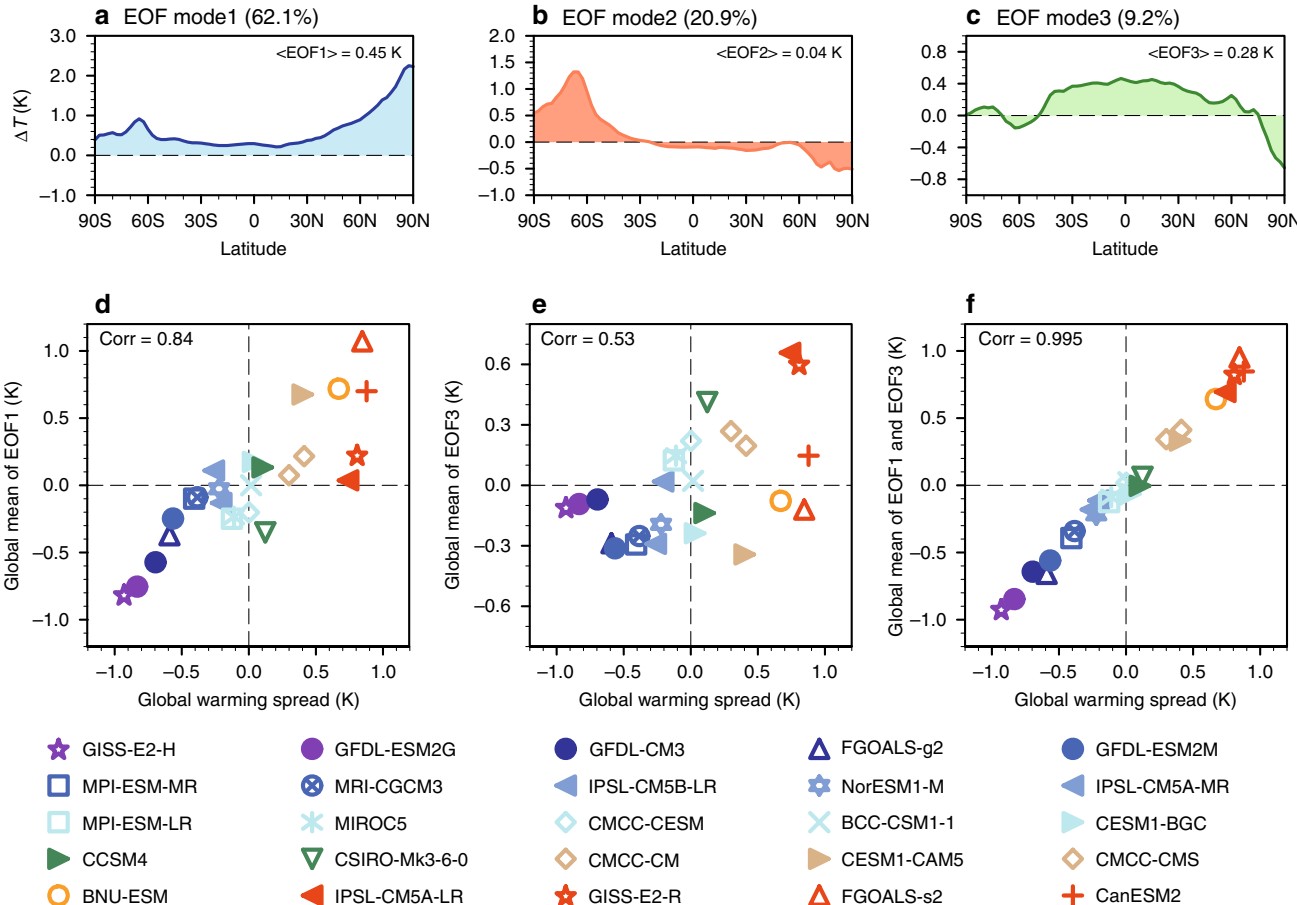

**Fig. 2 Dominant spatial structures of the inter-model warming pattern spread (WPS). a–c** The spatial patterns of the first three empirical orthogonal function (EOF) modes (*K*) that account for almost all of the zonal mean inter-model surface warming spread. **d–f** Scatter diagrams of model deviations from the global and ensemble mean surface temperature change, which is the global warming spread (GWS, *K*; abscissa) versus that captured by EOF1, EOF3, and their sum (*K*; ordinate). The numbers inside the parenthesis on top of **a–c** correspond to the percentage of the variance in the WPS explained by the first three EOF modes, while the numbers inside < > on **a–c** are their respective global-mean values (*K*). The inter-model GWS captured by an EOF model is determined from the product of the global-mean value of its spatial pattern and a principal component. The principal components of these three EOF modes are shown in Supplementary Fig. 3.

**Relationship between the GWS and WPS**. An EOF (empirical orthogonal function) analysis is a tool that can extract important spatial structures that account for as much of the variance in the inter-model WPS as possible. By projecting the inter-model spreads of feedbacks to the dominant spatial structures, we can isolate the processes that contribute to the inter-model WPS from those that do not. The EOF analysis reveals that the first three modes account for, respectively, 62.1%, 20.9%, and 9.2% (total of 92.2%) of the inter-model WPS (Fig. 2a–c; their principal components are shown in Supplementary Fig. 3). The global-mean warming spread captured by the first EOF mode (EOF1) explains about 70.6% of the variance in the GWS as indicated by their correlation of 0.84 (Fig. 2d), while EOF3 accounts for another 28.1% of the variance in the GWS (correlation = 0.53, Fig. 2e). Their sum has a nearly perfect correlation (about 0.995) with the GWS (Fig. 2f), indicating that 98.7% of the GWS is associated with the global means of these two dominant spatial patterns. The spatial pattern of EOF1 reveals that models with greater (smaller) global-mean warming have more (less) warming everywhere with stronger (weaker) PWA. The spatial pattern of EOF3 indicates that some models with greater (smaller) global-mean warming have stronger (weaker) warming in the tropics but with a less (more) pronounced PWA. The EOF2 mode represents models with enhanced (reduced) PWA in the southern hemisphere but reduced (enhanced) PWA in the northern

hemisphere. This opposing spatial pattern offsets in the global mean and therefore contributes little to the GWS. This highlights the additional insight gained with a zonal mean versus a global-mean analysis, as EOF2 is hidden in a global-mean analysis. Furthermore, the EOF analysis accentuates the connection between the spatial and global-mean analyses.

**Spread of climate feedbacks and WPS spatial patterns**. A regression analysis of the inter-model spreads of individual feedback processes against the principal components (Supplementary Fig. 3) of the first three EOF modes reveals that the inter-model spread of the ice-albedo feedback is the largest positive contributor to the WPS spatial patterns captured by EOF1 and EOF2, which is primarily responsible for the large regional WPS over the poles (Fig. 3). The inter-model spread of the water vapor feedback is the largest positive contributor to the WPS spatial pattern of EOF3 and the second largest positive contributor to EOF1. It follows that EOF1, the most dominant spatial pattern for the WPS and GWS, is primarily caused jointly by ice-albedo and water vapor feedbacks such that models with a greater (smaller) albedo feedback also have a larger (smaller) water vapor feedback. Their collective effect leads to a stronger (weaker) warming at all latitudes with a more (less) pronounced

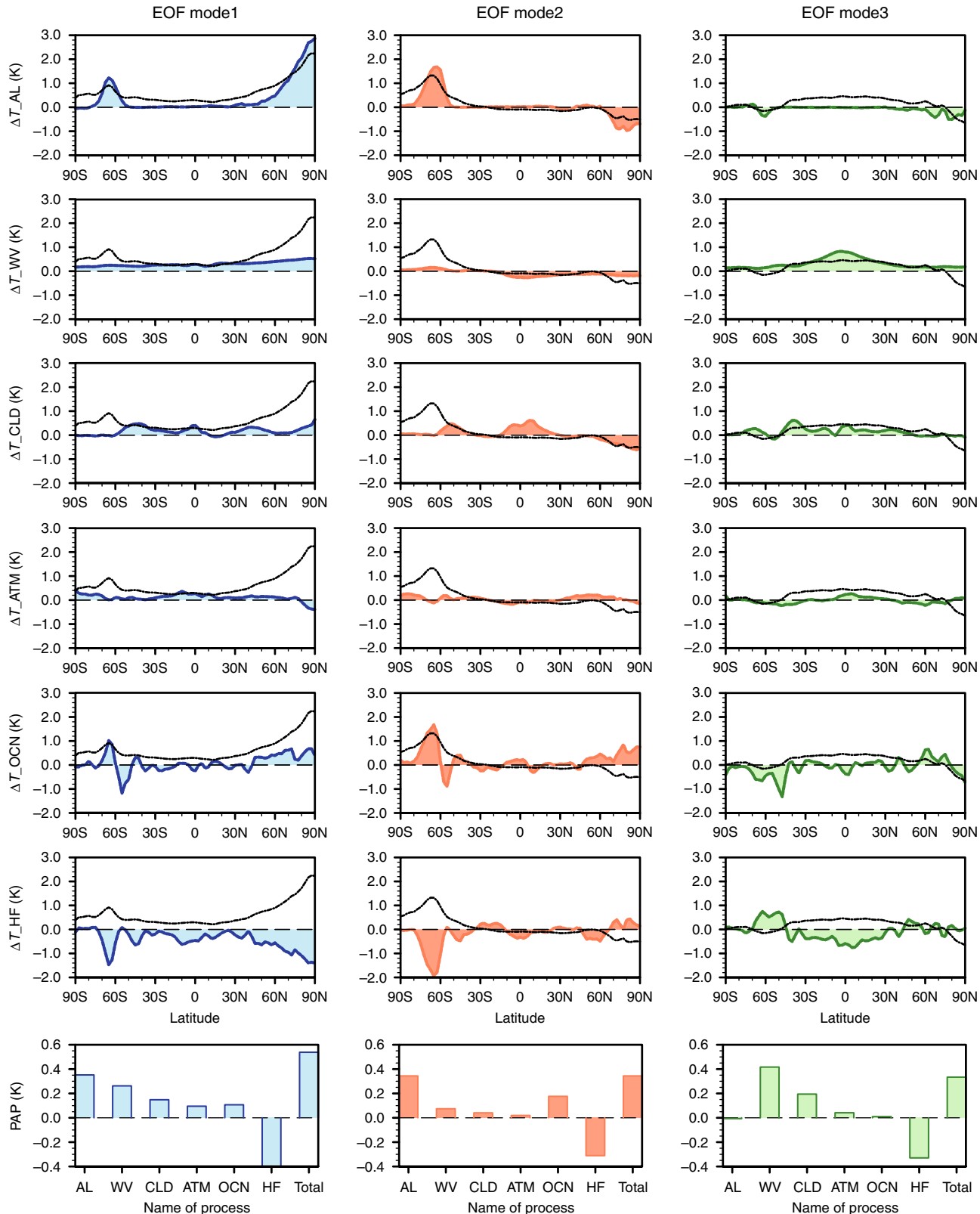

**Fig. 3 Individual process contributions to the three most dominant modes.** The regressed patterns (the top six rows) of the inter-model warming spreads in the individual (partial) surface temperature changes (Fig. 1e–j) against the principal components of EOF1 (empirical orthogonal function, the left column), EOF2 (the middle column), and EOF3 (the right column). The bottom row shows the pattern-amplitude projection (PAP) coefficients (*K*), measuring the contributions of individual processes to the first three EOF patterns whose sum (i.e., the total) gives the root mean square amplitude (*K*) of the corresponding EOF mode (see the Methods section for details). The abbreviations AL, WV, CLD, ATM, OCN, and HF stand for processes of ice-albedo, water vapor, clouds, atmospheric dynamics, ocean dynamics plus heat storage, and surface turbulent heat fluxes, respectively.

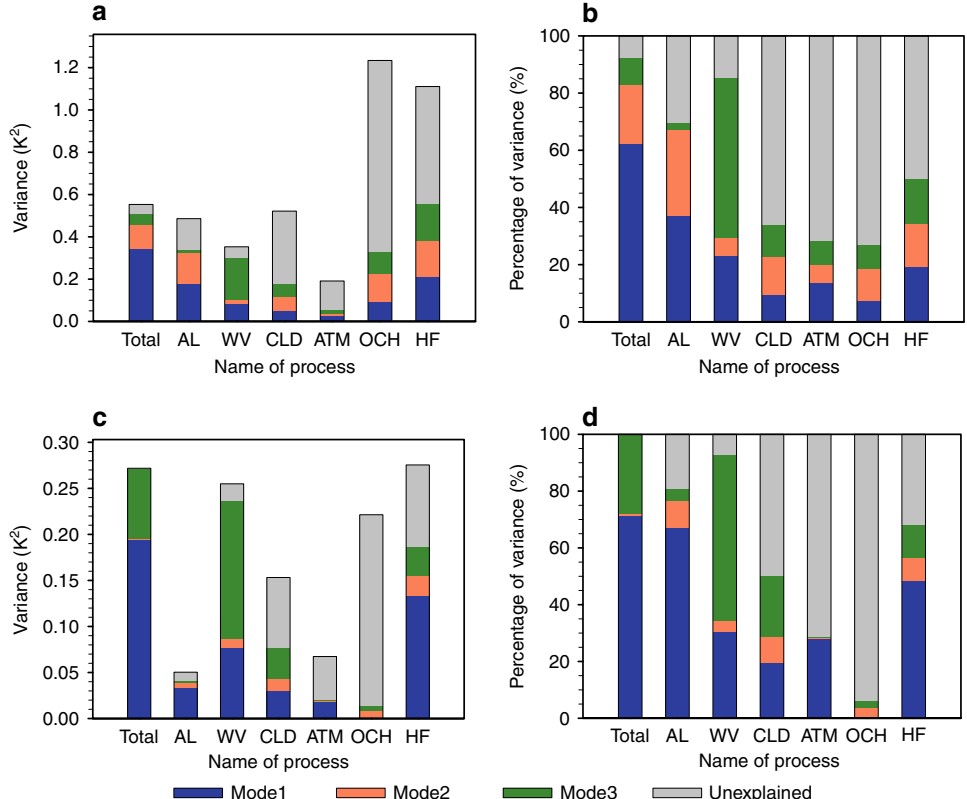

**Fig. 4 Variances of warming pattern spread (WPS) and global warming spread (GWS) explained by the three most dominant modes. a** The variance ($K^2$) of the zonal mean fields of the inter-model spread in total surface temperature change and the individual partial surface temperature changes. The color portions of each bar represent the variance explained by the first three EOFs (empirical orthogonal function), while the gray portion is the remaining variance. **b** The percentage (%) of the variance explained by the first three EOF modes (colors) and the remaining unexplained percentage by the three modes is in gray. **c**, **d** Same as **a** and **b** but for the variance of GWS. The abbreviations AL, WV, CLD, ATM, OCN, and HF stand for processes of ice-albedo, water vapor, clouds, atmospheric dynamics, ocean dynamics plus heat storage, and surface turbulent heat fluxes, respectively.

PWA. The dominant EOF1 mode and the positive correlation of the ice-albedo feedback in high latitudes and water vapor feedback in low latitudes can also be inferred from a regression analysis against the GWS (Supplementary Figs. 4 and 5). The spatial pattern of EOF3 also contributes substantially (28.1%) to the GWS and is mainly driven by the water vapor feedback spread. Models with a greater (smaller) water vapor feedback tend to have more (less) warming in the tropics relative to the warming in high latitudes, which contributes to a larger (weaker) global-mean warming.

Besides ice-albedo and water vapor feedbacks, another important contributor to the three dominant spatial patterns of the WPS is the inter-model spread in surface turbulent heat flux changes (the second row from the bottom in Fig. 3). However, unlike the ice-albedo and water vapor feedbacks, surface turbulent heat fluxes oppose each of the three dominant spatial patterns of the WPS. Such consistent opposition across all models indicates that the inter-model spread of surface turbulent fluxes acts to damp the inter-model spreads of ice-albedo and water vapor feedbacks throughout the model ensemble. In other words, models with larger (smaller) positive ice-albedo and water feedbacks tend to have a greater (lesser) increase in surface turbulent fluxes from the surface to the atmosphere. The offset, however, is only partial.

The other feedbacks, including the cloud feedback, contribute relatively little to the three dominant patterns of the WPS (Fig. 3), even though their individual spreads have a larger spatial variance than the inter-model spreads of ice-albedo and water vapor feedbacks (Fig. 4a). The weaker connections of the cloud and

remaining feedbacks to the spatial variance of the three dominant patterns of the WPS are depicted in Fig. 4a by the small colored portions of the variance. Focusing on the colored portions, the feedbacks with the largest explained variance of the WPS are the water vapor and ice-albedo feedbacks, plus changes in surface turbulent heat fluxes. The three dominant modes explain not only 92.2% of the total variance of the WPS, but also a majority of the spatial variance of water vapor and ice-albedo feedbacks (Fig. 4b). For the other feedbacks, the three dominant modes explain less than half of the variance in their respective spreads.

A breakdown of feedback contributions to the GWS provides a similar picture as the zonal mean analysis, but with some key subtle distinctions. The lack of an EOF2 contribution to the GWS is clearly depicted, as EOF1 and EOF3 almost fully explain the GWS (Fig. 4c). The magnitude of the global-mean albedo feedback's spread is much smaller, while the spreads in the global-mean water vapor feedback and changes in surface turbulent heat fluxes are elevated, since polar regions cover less area than the tropics. Nevertheless, a majority (more than 80%) of the spread in the global-mean water vapor and ice-albedo feedbacks is associated with the EOF1 and EOF3 modes (Fig. 4d). The results shown in Fig. 5a further illustrate the dominance of the positive contributions from ice-albedo and water vapor feedbacks and the suppression from the surface turbulent flux feedback perturbations that gives rise to the GWS of individual models. It is also seen from Fig. 5a that models that have larger global-mean warming tend to have a larger warming contribution from the cloud feedback, which explains the relatively high positive correlation of the temperature spread due to the cloud

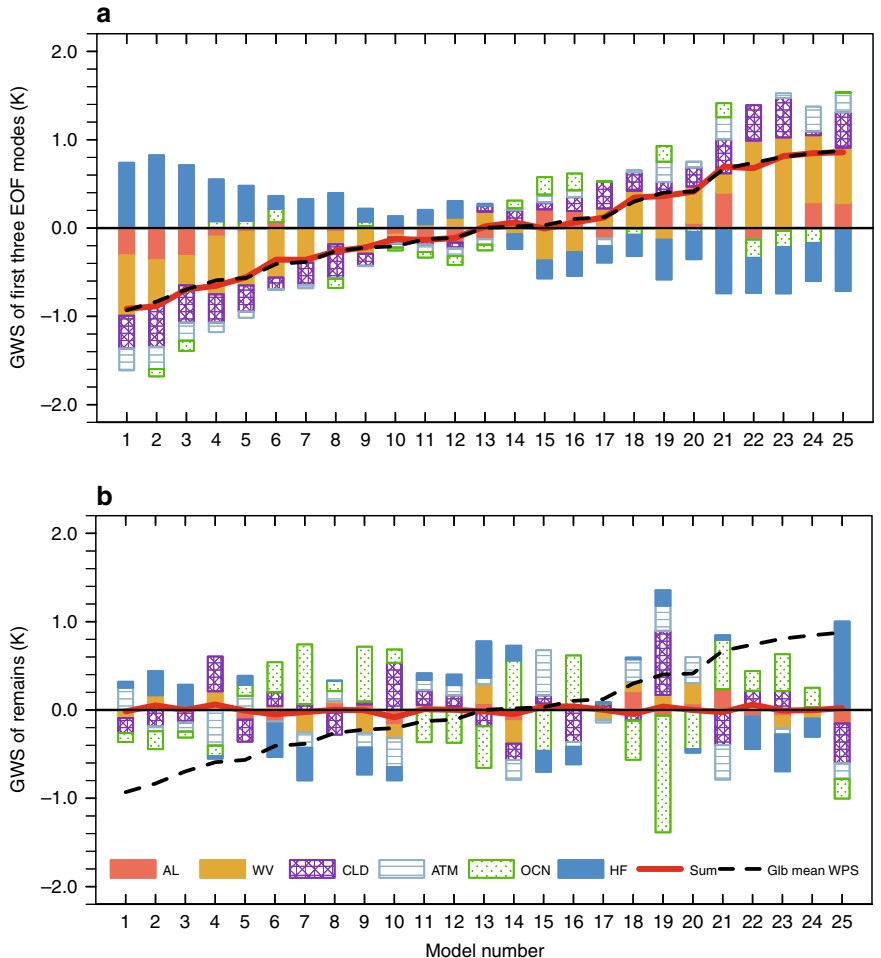

**Fig. 5 Stacked bar chart of the inter-model spreads of partial temperature changes. a** Global means of partial temperature changes shown in Fig. 1e–j (*K*; ordinate) as a function of model number (abscissa) captured by the first three EOF (empirical orthogonal function) modes and **b** the same as **a** but for the remaining fields. Different feedback processes are represented by different colors using the color scheme on the bottom. The solid red line corresponds to the (vertical) sum of the stacked color bars and the dashed black line corresponds to the global mean of individual models' temperature changes shown in Fig. 1c, namely, the global warming spread (GWS), or the global (glb) mean of warming pattern spread (WPS). Models are in the order of their projected global-mean surface warming from the smallest to the largest. The abbreviations AL, WV, CLD, ATM, OCN, and HF (f) stand for processes of ice-albedo, water vapor, clouds, atmospheric dynamics, ocean dynamics plus heat storage, and surface turbulent heat fluxes, respectively.

feedback with the GWS (Supplementary Fig. 2a). However, EOF1 and EOF3 still explain less than half of the variance in the cloud feedback (Fig. 4d).

**Comparison with the TOA perspective**. Aspects of these results appear at odds with previous studies. The source of the discrepancy could be attributed to the use of different methodologies, namely the use of a surface perspective instead of a TOA perspective and/or the use of spatial patterns to link the GWS to inter-model spreads of feedbacks. Shown in Fig. 6, Supplementary Figs. 7 and 8 are the TOA counterparts of Figs. 1, 3, and 4, respectively. The regression of the partial radiative flux perturbations (PRPs) produced by individual radiative feedbacks at the TOA against the three dominant EOF modes of the meridional surface warming pattern provides a similar picture to that derived from the surface perspective. Specifically, aside from the Planck feedback, the cloud feedback exhibits the largest inter-model spread in the zonal mean PRPs at the TOA (Fig. 6a and Supplementary Fig. 7). However, despite its large amplitude, the TOA cloud feedback spread projects relatively weakly onto the first three EOF modes (Fig. 6a and Supplementary Figure 8), such that

the first three EOF modes account for less than 40% of the variance in the cloud feedback. Consistent with the surface perspective, Fig. 6b indicates ~75% (~65%) of the zonal mean spread in the TOA radiative flux perturbations due to the water vapor (ice-albedo) feedback is linked to the first three EOF modes (less than 10% difference with its surface counterpart). Furthermore, Fig. 6 and Supplementary Fig. 8 show that both the water vapor and ice-albedo feedback spreads project strongly onto the first EOF mode, but only the ice-albedo (water vapor) feedback spread has a large projection onto the second (third) EOF mode, matching the surface perspective. The consistency between the two perspectives persists for the GWS (Fig. 4c, d versus Fig. 6c, d). As with the surface perspective, the cloud feedback uncertainty is a greater contributor to the GWS than the zonal mean WPS but still has a weaker link with EOF1 and EOF3 (Fig. 6c, d) than the water vapor feedback spread. Therefore, the difference between the TOA and surface perspectives does not account for the difference of our results with previous studies.

We have performed a similar analysis concerning the relation between the GWS and inter-model spreads of climate feedbacks but without considering their spatial patterns. The results shown in Supplementary Fig. 2 (surface perspective) and Supplementary

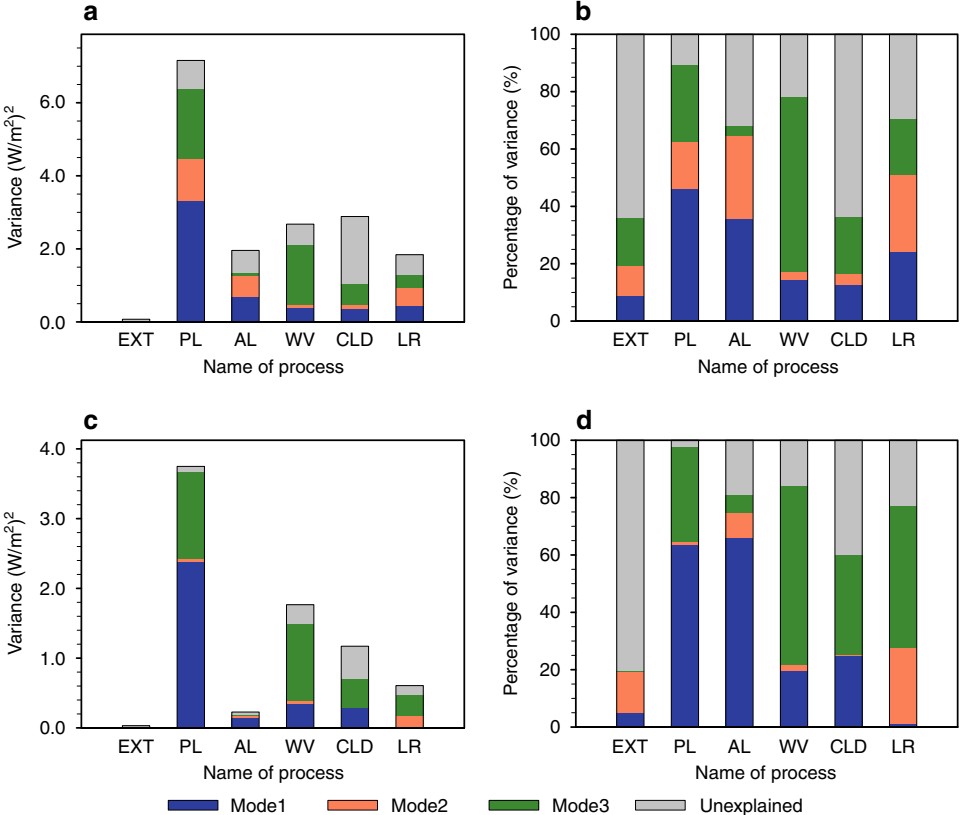

**Fig. 6 Variance of partial radiative perturbations (PRP) at the top of the atmosphere (TOA) explained by the three dominant modes. a** The variance (in units of $(W/m^2)^2$) of the zonal mean fields of inter-model spreads of PRPs at TOA. The color portions of each bar represent the variance explained by the first three EOFs (Empirical Orthogonal Function), while the gray portion is the remaining variance. **b** The percentage (%) of the variance explained by the first three EOF modes (colors) and the remaining unexplained percentage by the three modes is in gray. **c, d** Same as **a** and **b** but for the variance of their global means. The labels on the bottom of each panel stand for external forcing (EXT), Planck feedback (PL), surface albedo feedback (AL), water vapor feedback (WV), cloud feedback (CLD), and lapse-rate feedback (LR).

Fig. 9 (TOA perspective), which are obtained without regressing against the EOF modes, support the general consensus from a large number of previous studies, referenced above, the last three IPCC reports, as well as recent CMIP6 model results[25–27]; namely, the cloud feedback has a large inter-model spread and a strong correlation with the global-mean WPS. It is of interest to point out that when not normalizing by the total global-mean temperature change, both Supplementary Figs. 2 (surface) and 9a (TOA) reveal nearly equal important roles of the water vapor, clouds, and ice-albedo feedbacks for the GWS. In addition, there is little difference between the global-mean results obtained with and without considering the spatial patterns (e.g., Fig. 4c, d versus Supplementary Fig. 2 for the surface perspective and Fig. 6c, d versus Supplementary Fig. 9a for the TOA perspective). This is because the sum of the global means of the EOF1 and EOF3 modes explains nearly 99% of the GWS. It follows that with and without the consideration of the spatial pattern and both the TOA and surface perspectives would yield the same conclusion about the GWS, namely that the inter-model spreads of other feedbacks (e.g., water vapor and ice-albedo feedbacks) are as important as the cloud feedback in contributing to the GWS. However, the consideration of the inter-model spread in the meridional surface warming pattern reduces the cloud feedback's contribution to the GWS, relative to the water vapor and ice-albedo feedbacks (Fig. 4b versus Fig. 4d for the surface and Fig. 6b versus Fig. 6d for the TOA).

When using climate feedback parameters or normalizing the PRPs at the TOA for each model by their respective global-mean

surface temperature change, we can reproduce the results found in previous studies. Specifically, the inter-model spread of the cloud feedback parameter can be singled out as the main contributor, among all feedback parameters, to the GWS (Supplementary Fig. 9b). Therefore, there is no real discrepancy between our results and previous studies in terms of the GWS because it can be easily resolved when the PRPs at the TOA are normalized by the global-mean surface temperature change. Mathematically speaking, climate feedback parameters may provide a better measurement of the amplitude of the inter-model spread of feedbacks because the normalization by the global warming of individual models would factor out the portion of the feedback spread connected to the GWS. However, the correlation analysis between the inter-model spread of a climate feedback parameter (which is in units of $W/m^2/K$) and GWS is not the same as the correlation analysis between the feedback spread (which is in units of $W/m^2$) and GWS. Because climate feedback parameters are defined as feedbacks (in units of $W/m^2$) divided by the global-mean warming, the inter-model spread of a climate feedback parameter includes the information of both the inter-model spread in the feedback (in units of $W/m^2$) itself and the GWS. As a result, the correlation analysis between a climate feedback parameter and GWS automatically includes a built-in perfect negative correlation with the GWS, which in turn may compromise the correlation between the GWS and the inter-model spread of feedback processes (defined in units of $W/m^2$). One can easily prove the existence of such built-in perfect negative correlation by considering a special case in which the

feedback (in units of W/m$^2$) has a non-zero MME value but no inter-model spread. In this special case, the correlation between the inter-model spread of the climate feedback parameter and GWS is equal to −1 (i.e., the correlation between the GWS and its inverse), which would suggest a strong contribution of the feedback under the consideration to the GWS even though the lack of a spread in the feedback clearly indicates no connection to the GWS.

It is important to discuss the Planck and lapse-rate feedbacks that appear in the TOA perspective, but not in the CFRAM surface perspective because these feedbacks project strongly onto the first three EOF modes. By virtue of the definition of the Planck feedback, the spatial pattern of the inter-model spread of the Planck feedback can be regarded as the mirror image of the WPS. Thus, the regressed patterns of the Planck feedback against the first three EOF modes, after reversing their polarity, are nearly indistinguishable from the spatial patterns of the EOF modes of the WPS (second row of Supplementary Fig. 8 versus the top row of Fig. 2). By definition, the lapse-rate feedback tends to be positively (negatively) correlated with the Planck feedback (surface temperature change) when the air temperature warming is greater than the surface warming. Conversely, the lapse-rate feedback is negatively (positively) correlated with the Planck feedback (surface temperature change) when the air temperature warming is smaller than the surface warming. Because the air temperature warming is stronger (weaker) than the surface warming in the tropics (polar regions), it is expected that the inter-model spread of the lapse-rate feedback would have the same polarity with the Planck feedback in the tropics but opposite polarity in the polar regions, as evident by comparing the second row with the last row of Supplementary Fig. 8.

## Discussion

Removing the spread associated with the three dominant patterns, which account for 92.2% of variance of the WPS with their global means accounting for 98.7% of the GWS, reveals that the remaining feedback spreads offset each other through a model-dependent combination (Fig. 5b). Therefore, a majority of the cloud feedback spread is compensated by different model-dependent combinations of other feedback spreads that are also not well correlated with the three dominant patterns of the WPS. As a result, the inter-model spread of cloud feedback, despite its large amplitude, is not the leading contributor to the inter-model WPS and GWS. However, this result does not imply that cloud feedback is unimportant. Our results reveal that the coupling between the cloud feedback and other feedbacks varies from model to model. This suggests that the large spread in the cloud feedback could result from an inconsistent representation of the couplings between clouds and other processes among climate models. Such inconsistency could contribute to the inter-model WPS, although not directly, via their relationships to the feedback spreads of processes more responsible for the WPS (e.g., water vapor feedback). The inconsistent couplings between the cloud feedback and other feedbacks among models are troublesome, as it highlights discrepancies in model physics. Therefore, to reduce the inter-model spread in cloud feedback and better understand its role in establishing the WPS, it is important to study the couplings between clouds and other feedback processes in climate models and when possible in observations at climate change time scales (decades and longer). The same is true for the other processes that display inconsistent coupling (e.g., changes in ocean dynamics). Understanding the spatial coupling between feedbacks is critical to resolving the causes of the inter-model WPS.

Recent studies suggest clouds are responsible for the enhanced climate sensitivity in CMIP6 models[25–27]. The results of this study are not incongruous with these findings. Changes in model physics from CMIP5 to CMIP6 could impact the cloud feedback contribution to the climate sensitivity and the inter-model GWS. If the coupling between the cloud feedback and other feedbacks becomes more consistent among models such that models with larger cloud feedback consistently warm more, then the cloud feedback would increase the climate sensitivity. Moreover, if the cloud feedback spread remains large then it would also contribute more to the inter-model WPS and GWS.

An outstanding issue in the climate science is the large inter-model uncertainty of the surface warming amount in response to the same anthropogenic greenhouse radiative forcing. Here we identify two dominant spatial patterns, driven mainly by the inter-model spreads of ice-albedo and water vapor feedbacks, that account for 71.3% of the inter-model spread in the spatial warming pattern and 98.7% of the spread in global-mean warming projections under the CMIP5 RCP8.5 scenario. We further reveal the cloud feedback spread, despite its large amplitude, contributes less to the inter-model spread of global warming projections because it tends to be canceled by other feedbacks that are also not well correlated with these two dominant spatial patterns. The findings of this study indicate that the large WPS and GWS can be significantly reduced by narrowing the inter-model spread of ice-albedo and water vapor feedbacks.

## Methods

**Data.** Supplementary Table 1 lists the 25 models' RCP8.5 simulations considered in this study in the order of their global-mean surface temperature changes from the smallest warming to the largest warming, whereas Supplementary Table 2 lists the 7 models whose CMIP5 simulations are archived in the CMIP5 websites but are not used in this study because of the lack of either cloud fields or the information necessary to process cloud fields. The historical and future climate states are evaluated as the 50-year means of the 1951–2000 period in the historical simulations and the 2051–2100 period in the RCP8.5 simulations. The surface temperature difference between the future and historical climate states of each model corresponds to the global warming projection of that model.

**Feedback calculations.** Following Hu et al.[3], we have applied the climate feedback-response analysis method (CFRAM)[23,24] to calculate partial surface temperature changes at each location due to the external forcing alone and due to each of the individual feedbacks alone. Supplementary Table 3 lists the specific output fields of the historical and RCP8.5 simulations that are actually used in the CFRAM feedback analysis. We perform the CFRAM calculations using the 50-year mean fields for the present and future climate states derived from the outputs of individual models with their own spatial (horizontal and vertical) resolutions (i.e., without any interpolations). The feedback processes considered include water vapor, ice-albedo, and clouds feedbacks, convective and advective energy transport by atmospheric dynamics, surface turbulent sensible and latent heat fluxes, oceanic energy transport and heat storage, and ozone. Note that in the framework of the CFRAM analysis, the effect of the lapse rate or air temperature feedback has been included in these partial surface temperatures[18], because changes in both air and surface temperatures are considered to be the response to changes in the external forcing and individual feedbacks.

We produce a total of ten global maps for each model, eight of which correspond to the partial temperature changes due to the (i) external forcing alone, (ii)–(viii) each of the seven aforementioned feedbacks, while the remaining two correspond to the (ix) (total) surface temperature change derived from the original CMIP5 climate simulations, and (x) the difference between the model-derived surface temperature change and the sum of all partial surface temperature changes obtained from the CFRAM feedback calculations. The smallness of the latter is indicative of the accuracy and robustness of the CFRAM feedback calculations (Supplementary Fig. 1j) because it measures the closeness of the sum of the partial surface temperature changes to the (total) surface temperature change derived from the original climate simulations.

Beside these partial surface temperature changes, we have also calculated the partial radiative flux perturbations (PRPs) at the TOA due to the external forcing and due to the Planck, water vapor, ice-albedo, cloud, and lapse-rate feedbacks.

To facilitate the statistical analysis of the ten surface maps and the six TOA fields across the 25 models, we have interpolated them onto a common grid with a 1°×1° horizontal resolution. The results of the statistical analysis of these ten surface maps and the six TOA fields across the 25 models can be adequately represented by the results obtained from their zonal mean fields. To make the graphic presentation manageable, we will only present the results obtained from the zonal mean fields (unless specified otherwise).

**Statistics analysis procedures**. We consider an individual models' surface warming deviation from the 25 multi-model ensemble (MME) mean warming projection as a measure of the global warming spread (GWS) or meridional warming pattern spread (WPS). Supplementary Table 1 summarizes global-mean warming projections and their deviations from the MME warming projection. Similarly, we obtained the spreads of the partial temperature changes associated with individual processes by removing their respective MME mean values. As indicated in Fig. 1 and Supplementary Fig. 1, only six partial temperature changes, namely those due to water vapor, ice-albedo, and clouds feedbacks, convective and advective energy transport by atmospheric dynamics, surface turbulent sensible and latent heat fluxes, and oceanic energy transport and heat storage, exhibit large spreads. There is little spread in the partial temperature changes due to the external forcing alone and ozone, as well as in the error term of the CFRAM analysis. Therefore, our attribution of the sources of the GWS mainly focuses on the six feedback processes that display large inter-model spreads.

Our attribution of the sources of the GWS begins with an EOF (Empirical Orthogonal Function) analysis of the inter-model WPS. The latitudinal variations of the inter-model WPS are represented by individual EOF modes whose amplitude (measured by the spatial variance) indicate their relative importance (or the variance explained). The (normalized) principal components of individual EOF modes are the projections of spreads of individual models' warming projections from the MME warming projection. We have also applied the EOF analysis to the two-dimensional deviations of individual model surface warming projections from the MME average. The results remain qualitatively unchanged except the total variance explained by the first three modes is less (68.3% instead of 92.2%; Supplementary Fig. 6). The zonal mean of the two-dimensional EOF modes resembles the zonal mean EOF modes except that the order of the two-dimensional EOF2 and EOF3 modes are reversed. Therefore, the conclusions made based on the zonal mean EOF analysis hold.

We use the principal components of the individual EOF modes to regress the zonal mean spreads of individual feedback processes to identify the spatial patterns of individual feedback processes associated with the individual EOF modes of the inter-model WPS. The amplitude of the regressed spatial patterns, both in terms of the absolute amplitude (spatial variance of feedback spreads) and relative amplitude (explained variance of individual feedback spreads), is indicative of the importance of individual feedback processes to the individual EOF modes and the (total) inter-model WPS. Following Deng et al.[28], we also use the pattern-amplitude projection (PAP) to succinctly measure the individual feedback contributions to individual EOF modes of the inter-model WPS, according to

$$\text{PAP}_j^{(X)} = \frac{\int_{-\pi/2}^{\pi/2} \Delta^{(x)} T_j \Delta T_j \cos\phi \, d\phi}{\sqrt{\int_{-\pi/2}^{\pi/2} (\Delta T_j)^2 \cos\phi \, d\phi}}, \tag{1}$$

where $\phi$ is latitude, $\Delta T_j$ is the $j$th EOF mode of the inter-model WPS, and $\Delta^{(X)} T_j$ is the regressed pattern of the partial temperature change due to $X$ process against the $j$th principal component. Because the sum of $\Delta^{(X)} T_j$ over $X$ is convergent to $\Delta T_j$ (due to the robustness of the CFRAM analysis), the sum of $\text{PAP}_j^{(X)}$ is equal to the root mean square of the $j$th EOF mode of the inter-model WPS.

## Data availability

All data used in this study are derived from the monthly mean outputs of the Historical and RCP8.5 (Representative Concentration Pathway) model simulations produced by the Coupled Model Intercomparison Project Version 5 (CMIP5), which are archived and freely accessible at http://data.ceda.ac.uk/badc/cmip5/data/cmip5/output1 and https://esgf-node.llnl.gov/search/cmip5/. The data generated from our calculations are available on request from authors. Source data, model variables, definitions, and extended data display items are available in the online version of the paper, and references unique to these sections appear only in the online paper.

## Code availability

The codes used for our calculations are available on request from authors.

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

## Acknowledgements

This research was in part supported by grants from the National Natural Science Foundation of China (41805050, 41690123), National Science Foundation (AGS-1354834 and AGS-2012479), NASA Interdisciplinary Studies Program Grant NNH12ZDA001N-IDS, and NASA Postdoctoral Program administered by Universities Space Research Association.

## Author contributions

M.C. conceived the idea for the study; X.H. and H.F downloaded the data and performed the calculations. All authors discussed the results throughout the whole process; M.C., S. S., P.T., and X.H. were the main writers, and H.F. and S.Y. contributed to the writing. X. H. and H.F. are co-first authors.

## Competing interests

The authors declare no competing interests.
