## [Peer Review File · Nature Communications]

Response to Reviewer 3:

Reviewer #3 (Remarks to the Author):

I find my comments overall satisfactorily addressed, and therefore I recommend the paper for publication after addressing a few remaining points below. I think that it now reads much more clearly than it did earlier, and should be a worthwhile contribution to the literature.

I have a number of remaining comments, typos or minor clarifications. The line numbers refer to the marked up version of the manuscript, as I found this easier to use in the review process:

Response: Thanks for the third round of reviewing our manuscript and these specific suggestions and corrections that help improve the clarity of the presentation greatly. Below are our point-to-point responses with the line numbers referring to the clean version of the manuscript.

(1): line 96: I think it is important to state more clearly that the authors are using the term "climate feedback parameters" to define what is normally referred to in the literature as "climate feedbacks", given that many readers will not immediately pick up this terminology. So a few extra words and emphasis (in italics) here would help e.g.

"moreover, the majority of work in the literature focuses on what we term here as "climate feedback parameters", viz TOA radiative flux changes associated with individual feedback processes, *normalised by global mean temperature change*. By contrast, here we adopt..."

Response: We believe that the climate science community does use the term "climate feedback parameters (almost exclusively) for climate feedback processes (e.g., see the seminal review paper by Bony et al. 2006). In our work, we consider "*global means of partial energy flux perturbations at the TOA associated with internal processes or their equivalent at the surface by converting partial energy flux perturbations at the surface to partial surface temperature changes, without any normalization*" (Lines 80-82). We believe that our statements about the terminology used in the literature are correct and therefore, no changes have been made here.

Bony, S. *et al.* How Well Do We Understand and Evaluate Climate Change Feedback Processes? *J. Clim.* **19**, 3445–3482 (2006).

(2): there needs to be more discussion amplifying the sentence starting on line 308: "However, feedback parameters may not be suitable for the correlation analysis because by virtue of the definition of climate feedback parameters, such correlation analysis automatically includes a built-in perfect negative correlation with the GWS, which in turn may compromise the true correlation between the GWS and the inter-model spread of feedback processes."

Specifically what evidence is there that there has been a "built-in perfect negative correlation" between traditionally defined feedbacks and global warming spread? Analyses of feedbacks and global temperature spread don't normally show extremely high correlations, perhaps with the exception of clouds, where there is a strong positive correlation. E.g. there is no negative correlation between say surface albedo feedback (as traditionally defined in the literature) and the spread in global temperature between models.

Can the authors also please make the sentence itself clearer, specifically

- which correlation analysis are they talking about?
- What you mean by the "true" correlation?
- Incidentally, I would drop "by virtue of the definition of climate feedback parameters" for brevity and readability.

Response: In the revision, we have provided more elaborative discussions on the difference between the two correlation analyses. We have provided detailed information about the (mathematical) evidence for the existence of a built-in perfect negative correlation between a climate feedback parameter and GWS. We deleted the word “true” and the phrase “by virtue of the definition of climate feedback parameters”. Now these statements (Lines 240-255) read as

“However, the correlation analysis between the inter-model spread of a climate feedback parameter (which is in units of $W/m^2/K$) and GWS is not the same as the correlation analysis between the feedback spread (which is in units of W/m^2) and GWS. Because climate feedback parameters are defined as feedbacks (in units of W/m^2) divided by the global mean warming, the inter-model spread of a climate feedback parameter includes the information of both the inter-model spread in the feedback (in units of W/m^2) itself and the GWS. As a result, the correlation analysis between a climate feedback parameter and GWS automatically includes a built-in perfect negative correlation with the GWS, which in turn may compromise the correlation between the GWS and the inter-model spread of feedback processes (defined in units of W/m^2). One can easily prove the existence of such built-in perfect negative correlation by considering a special case in which the feedback (in units of W/m^2) has a non-zero MME value but no inter-model spread. In this special case, the correlation between the inter-model spread of the climate feedback parameter and GWS is equal to -1 (i.e., the correlation between the GWS and its inverse), which would suggest a strong contribution of the feedback under the consideration to the GWS even though the lack of a spread in the feedback clearly indicates no connection to the GWS.”

(3) line 103: "equivalence" should read "equivalents", when referring to the TOA versus surface correspondence

Response: We have replaced “equivalence” with “equivalents” (Line 81).

(4): line 292: "nearly explains 99%" should read "explains nearly 99%"

Response: It has been changed per the suggestion (Line 224).

(5): sentence starting line 305: remove the second "mathematically" from the sentence for readability.

Response: The second “mathematically” has been deleted (Line 239).

Thank you also for responding explicitly to each of my 15 minor comments.

Again, we are grateful for your carefully reading our manuscripts and these insightful and constructive comments/suggestions that greatly improve the presentation and readability of the manuscript.

Reviewers' comments, third round:

Reviewer #2 (Remarks to the Author):

The authors have addressed all my comments satisfactorily.

Reviewer #3 (Remarks to the Author):

I find my comments overall satisfactorily addressed, and therefore I recommend the paper for publication after addressing a few remaining points below. I think that it now reads much more clearly than it did earlier, and should be a worthwhile contribution to the literature.

I have a number of remaining comments, typos or minor clarifications. The line numbers refer to the marked up version of the manuscript, as I found this easier to use in the review process:

(1): line 96: I think it is important to state more clearly that the authors are using the term "climate feedback parameters" to define what is normally referred to in the literature as "climate feedbacks", given that many readers will not immediately pick up this terminology. So a few extra words and emphasis (in italics) here would help e.g.

"moreover, the majority of work in the literature focuses on what we term here as "climate feedback parameters", viz TOA radiative flux changes associated with individual feedback processes, *normalised by global mean temperature change*. By contrast, here we adopt..."

(2): there needs to be more discussion amplifying the sentence starting on line 308: "However, feedback parameters may not be suitable for the correlation analysis because by virtue of the definition of climate feedback parameters, such correlation analysis automatically includes a built-in perfect negative correlation with the GWS, which in turn may compromise the true correlation between the GWS and the inter-model spread of feedback processes."

Specifically what evidence is there that there has been a "built-in perfect negative correlation" between traditionally defined feedbacks and global warming spread? Analyses of feedbacks and global temperature spread don't normally show extremely high correlations, perhaps with the exception of clouds, where there is a strong positive correlation. E.g. there is no negative correlation between say surface albedo feedback (as traditionally defined in the literature) and the spread in global temperature between models.

Can the authors also please make the sentence itself clearer, specifically

- which correlation analysis are they talking about?
- What you mean by the "true" correlation?
- Incidentally, I would drop "by virtue of the definition of climate feedback parameters" for brevity and readability.

(3) line 103: "equivalence" should read "equivalents", when referring to the TOA versus surface correspondence

(4): line 292: "nearly explains 99%" should read "explains nearly 99%"

(5): sentence starting line 305: remove the second "mathematically" from the sentence for readability.

Thank you also for responding explicitly to each of my 15 minor comments.

Response to Reviewer 1:

Reviewer #1 (Remarks to the Author):

This is a solid study and ready for publication. The authors considered the reviews seriously and revised the manuscript with additional analyses presented in response. The three reviews were very similar and I feel that the authors did a good job at answering concerns regarding the role and relationship of cloud feedbacks in the study and the surface-TOA differences. The manuscript as it is now provides the reader with adequate information to interpret and further explore the implications for climate models.

One typo on L 183 - 'differently' should be 'different'

Response: Thanks for the second round of reviewing our manuscript. The typo has been corrected.

Response to Reviewer 2:

Reviewer #2 (Remarks to the Author):

I see that the Editor has decided to go forward with potential publication in this non-specialized journal despite my advice of the contrary. I think the study in general has merit, even if it uses the out-dated climate model ensemble. The authors have done well in addressing my key comments.

Below I now only list a few specific remarks that I recommend should be addressed before publication.

Response: Thanks for the second round of reviewing our manuscript. We have addressed these specific comments in the revision. Below are our point to point responses.

154-56: The general idea is that regional temperature changes approximately scale with the global-mean warming. I don't think the authors invalidate this hypothesis. As such, the first sentence needs to be revised, more clear, or dropped. The second sentence in fact implies Arctic warming scales with the global mean one, which implies the former sentence (154 – 155) is wrong.

Response: We have deleted that sentence in the revision. The first four sentences have been rewritten as *“The GWS is accompanied by the inter-model spread of geographic warming patterns, as the amount of warming varies greatly from location to location. Arctic surface warming, for example, is three to four times greater than the global mean warming^{2,11-12}. For an easy reference, we refer to the inter-model spread of geographic warming patterns as the “warming pattern spread” (WPS). The central question of this study is to explore if the GWS can be explained by a few dominant patterns of the WPS. If so, the task of identifying the factors giving rise to the GWS becomes revealing the feedback processes that are responsible for the WPS. There are a few studies relating the spatial patterns of the inter-model WPS to changes in clouds, albedo, water vapor, and other factors¹³⁻¹⁴”*.

168-70: This is an obscure use of the term “feedback”. A feedback in general is defined as a process that changes as a function of the global-mean surface temperature, and itself impacts the top-of-atmosphere radiation balance. I would guess a non-radiative feedback is a biogeochemical feedback (carbon cycle), but the keywords “ocean”, “atmospheric dynamics” or “surface turbulent heat fluxes” do not resonate with any pre-described feedback process. A very careful rewording is necessary that explains exactly what the authors have in mind.

Response: We replace the phrase “non-radiative feedbacks” with “non-radiative processes”.

177: What do the authors mean by “magnitude of the ... pattern”? is it the amplitude, i.e. difference between least and most warming regions, or is it the magnitude, i.e. the overall amount of warming?

Response: We meant the amount of warming (i.e., the magnitude). For clarity, this sentence now reads as *“Though this characteristic pattern persists across the CMIP5 (Coupled Model Intercomparison Project Phase 5) model ensemble, both the global mean and regional warming amounts exhibit substantial differences across models, particularly in polar regions (Fig. 1b)”*.

178-81: Fig. 1 suggests the authors think of the WPS as zonal, temporal mean meridional distribution of the warming. If so, this would be good to clarify at this point. If not, a clarification would help even more.

Response: To differentiate the spatial warming pattern from the global mean warming, we use “GWS” (global warming spread) to infer the inter-model spread of the global mean warming projection and “WPS” (warming pattern spread) to infer the inter-model spread of the meridional pattern of the warming projection. We have defined them clearly in Introduction.

Response to Reviewer 3:

Reviewer #3 (Remarks to the Author):

Re-review of A Less Cloudy Picture of the Inter-Model Spread in Future Global Warming Projections, by X. Hu et al.

Many/most of my queries have been answered by the authors. I think that the results that they show are interesting and quite important ones in that they shed light on the causes of different warming pattern changes in GCMs. This is an important new perspective on the role of feedbacks and climate change. They also shed light on some differences between surface and top of atmosphere changes and feedbacks. The extension of the analysis to the top of atmosphere is a very nice addition. However, I still find the way that they present the results confusing and think that upfront clarification and then wording changes throughout would clarify their presentation of results, make it much easier for the reader to understand and make their points of difference from earlier studies easier to understand.

Response: Thanks for the second round of reviewing our manuscript and these specific suggestions that help improve the clarity of the presentation greatly.

1) My understanding of the paper is as follows: The key point of the paper and the way it differs from previous studies on the importance of feedbacks in model projection spread is twofold. Firstly the present study considers a surface perspective for evaluating feedbacks in models rather than the conventional top of atmosphere approach. Secondly, it focuses primarily on the impact of different feedbacks on the pattern of surface temperature change, rather than on the spread in global mean warming.

This needs to be stated very clearly upfront. It could be better to define WPS as "warming pattern spread", rather than "warming projection spread" as this makes it clearer.?. Secondly when global temperature changes are being considered I strongly suggest you refer to them as "global temperature spread (GTS)", to differentiate the pattern from the global values discussion.

Response: Following your suggestions, we use "GWS" (global warming spread) to infer the inter-model spread of the global mean warming projection and "WPS" (warming pattern spread) to infer the inter-model spread of the meridional pattern of the warming projection. This indeed helps differentiate the pattern from the global values discussions. We have defined them clearly in Introduction (Line 48-53 and 56-69) and in lines 90-92:

"A quantitative evaluation of the physical mechanisms responsible for the inter-model differences in the global mean warming projection (referred to as "global warming spread" and abbreviated as GWS hereafter) is key to understanding why models exhibit varying degrees of warming when forced by the same increase of greenhouse gases. In the literature, the inter-model differences in cloud feedback strength has been argued as the main cause of the GWS⁴⁻⁹."

"The GWS is accompanied by the inter-model spread of geographic warming patterns, as the amount of warming varies greatly from location to location. Arctic surface warming, for example, is three to four times greater than the global mean warming^{2,11-12}. For an easy reference, we refer to the inter-model spread of geographic warming patterns as the "warming pattern spread" (WPS)."

“We define the WPS as the departures of individual models’ zonal mean temperature changes from the MME (Fig. 1c). The global mean of the WPS corresponds to the GWS”.

2) Abstract:

Therefore, I suggest in the abstract that the authors insert texts such as.

Both the global magnitude of the warming, and the geographical pattern of that warming however differ substantially between models. This paper focuses on feedbacks that control the pattern of warming. We refer to differences across models in their pattern of warming as "warming projection (or pattern) spread" (WPS). WPS limits our ability to foresee the severity of regional impacts on nature and society.

Further down in the abstract I don't understand what is meant by two patterns of warming explaining 97% of global temperature spread. Don't you mean 90% of the meridional pattern spread?

Response: We have rewritten the second to fourth sentences of Abstract (Lines 24-28) according to your suggestions. Now they read as *“Both the global mean warming, and the geographical warming pattern however differ substantially between models. This paper focuses on feedbacks that control the inter-model differences in their warming patterns or the "warming pattern spread" (WPS). WPS limits our ability to foresee the severity of regional impacts on nature and society”.*

We also revised the next sentence (Lines 28-30) as *“This study identifies two dominant patterns of WPS whose global mean values explain 98.7% of the global warming spread (GWS) of 2.6-4.0 K under the RCP8.5 ...”*, where the number 98.7% is obtained from the correlation of 0.995 shown in Fig. 2f (whose square is equal to 0.99) between the global mean values of the two patterns (ordinate of Fig. 2f) and the global mean warming of individual models (abscissa of Fig. 2f).

3) Introduction:

the first paragraph is talking about purely global temperature spread. Therefore drop all references in this paragraph to WPS, and change it to "global warming spread (GWS)" or something similar. There is no reference to patterns here therefore use of WPS is confusing.

Response: Thanks for suggesting the phrase “global warming spread” (GWS) to differentiate from the “warming pattern spread” (WPS). In the revision, we only reference to GWS in the first paragraph and we introduce WPS in the second paragraph.

4) Throughout the paper, whenever WPS refers to global changes, e.g. "global mean WPS", I suggest you change it to GWS. So for example, in the first line of the second paragraph change WPS to GWS.

Response: We have made such changes throughout the paper.

5) Further down there seems to be a second definition of WPS: the text says "We define the inter-model WPS as the range in model simulated global mean warming after removing the MME average (Fig. 1c). For easy reference, we call the departures of individual models’ temperature changes from the MME as the WPS."

This is very confusing: why is the WPS now defined as global mean warming, rather than the pattern? Incidentally figure 1C shows meridional changes, not global changes.

Response: We have rewritten these two sentences in Lines 90-92, which now read as “*We define the WPS as the departures of individual models’ zonal mean temperature changes from the MME (Fig. 1c). The global mean of the WPS corresponds to the GWS*”.

6) It is pleasing that the authors have now performed further calculations normalising the radiative changes by the global temperature change (lines 206-231) I have a number of comments about this specifically:

a. firstly, it deserves its own section. It sits in the section "top of atmosphere versus surface" but it is not about that surface/TOA issue. Rather it is about the issue of expressing changes as feedbacks, rather than absolute flux changes.

Response: There are three paragraphs in the section “Comparison with the TOA perspective”. The first one is about the pattern comparison of the TOA perspective (discussed in this section) with the surface perspective (which has already been discussed in the early part of the paper).

The second paragraph is devoted to the global mean analysis, which show the results “*in line with results elsewhere and e.g. supported in the last 3 IPCC reports*” when the global means of partial radiative flux perturbation at the TOA due to individual feedbacks are normalized by the global mean warmings. The third paragraph discusses the terms that do not have direct correspondence in the surface energy balance equation. Therefore, they are indeed all about the surface/TOA issue except the second paragraph focuses on the GWS and the first focus on the WPS.

We will respond to the comment “*Rather it is about the issue of expressing changes as feedbacks, rather than absolute flux changes*” in conjunction with the next comment below.

b. Secondly, the authors need to make explicit why normalising by surface temperature changes suddenly brings their results in line with results elsewhere and e.g. supported in the last 3 IPCC reports. It is not enough to just say "normalising by surface temperature make this happen" they need to say what is the essential difference now in the analysis when normalised by surface temperature that now makes the methodologies agree. I suspect it is because normalisation expresses the CFRAM flux changes instead as true feedbacks ($\text{W/m}^2/\text{K}$) rather than simply as flux changes (W/m^2). It could be argued the former are the cause of the global temperature spread, the latter are the result of the global spread. The authors need to discuss this.

7) Section starting line 181:

Related to the point above: when discussing the discrepancy with other results, one important issue is that the present study is looking at absolute changes in the surface fluxes which correspond with the spread in temperature between models. Although the authors use the word "feedbacks" to describe processes like surface albedo and water vapour responses they are not really feedbacks in the way that it is defined elsewhere in the literature, where radiative changes are normalised by global surface temperature changes ($\text{W/m}^2/\text{K}$). Therefore the authors need to specifically clarify in the text up front somewhere that where they refer to "feedbacks", this is shorthand for the net flux changes due to particular processes.

Response: We here take the liberty to respond to the comments #6b) and #7) jointly as they are highly related.

First, we wish to state that we do follow the standard textbook definition of “feedback”, which is defined as a change of energy input/output resulting from an internal process that in turn either amplifies or opposes the initial perturbation in energy input/output caused by the external forcing (i.e., both external forcing and feedback are measured as energy flux perturbations). Therefore, it is standard practice in the literature to use “feedbacks” as “a shorthand for the net flux changes due to particular processes”. The quantity of “radiative changes normalized by global surface temperature changes ($W/m^2/K$)” is referred to as **climate feedback parameter** in the literature (rather than just “climate feedback”). In the revision, we explicitly acknowledge that in the literature, majority of the work focus on climate feedback parameters (radiative changes normalized by global mean temperature change) in searching for the main sources of the climate projection uncertainty using climate models, but we focus on the direct comparison the GWS and inter-model spreads of feedback processes without normalization. Specifically, in the end of Introduction, we have added the following statements (Lines 74-84):

“Moreover, the majority of work in the literature focus on climate feedback parameters, which are defined as the TOA radiative flux changes associated with individual feedback processes normalized by global mean temperature change, in searching for the main sources of the global mean climate projection uncertainty using climate models. We here adopt the standard textbook definition of “feedback”, which is defined as a change of energy input/output resulting from an internal process that in turn either amplifies or opposes the initial perturbation in energy input/output caused by the external forcing. We will directly compare the WPS and GWS with inter-model spreads of spatial patterns and global means of partial energy flux perturbations at the TOA associated with internal processes or their equivalence at the surface by converting partial energy flux perturbations at the surface to partial surface temperature changes, without any normalization”.

Second, we should stress that the difference mentioned in the comment has nothing to do with methodologies, namely CFRAM and PRP methods. The results presented in the section “Comparison with the TOA perspective” are obtained by the standard PRP method (other than using different radiative transfer models). We have performed the same analysis from both the surface and TOA perspectives and examined the differences between the two perspectives finding that the two perspectives lead the same overall conclusions. By confirming this, we effectively rule out the possibility the seeming contradiction is due to the choice of the surface perspective. In this study, we further find one of the two sources for the difference between our findings and the others is due to the choice of the “absolute changes” (ours) versus the “relative changes” (i.e., normalized changes by the global mean temperature change or climate feedback parameters, which is considered by majority of studies in the literature). In other words, **we find that the “well-adopted consensus” that the inter-model spread of the cloud feedback is the largest source for the climate change projection uncertainty, is sensitive to the metric used to define feedback strength** (i.e., partial radiative energy flux/temperature perturbations vs. climate feedback parameters). The other source for the difference that we find with previous work is due to our novel consideration of the spatial pattern spreads of feedback processes.

We understand that normalization by global mean surface temperature provides a way to intercompare the contributions of the various feedbacks in a relative sense across models to assess what relative importance of each feedback to the warming in each model per K. Since the focus of this paper is not on this question of the “best metric” for feedback strength, we think that extensive discussion of this topic is beyond the scope of this manuscript. Nevertheless, we provide the following statements in the second paragraph of this section (Lines 234-241) to state our thoughts on the question of “relative changes” versus “absolute changes” on the basis of our results obtained by comparing the GWS with inter-model spreads of feedback processes with/without normalization. We hope this would help stimulate further discussions in the research community on this.

“Mathematically speaking, climate feedback parameters may provide a better measurement of the amplitude of the inter-model spread of feedbacks because the normalization by the global warming of individual models would mathematically factor out the portion of the feedback spread due to the GWS. However, feedback parameters may not be suitable for the correlation analysis because by virtue of the definition of climate feedback parameters, such correlation analysis automatically includes a built-in perfect negative correlation with the GWS, which in turn may compromise the true correlation between the GWS and the inter-model spread of feedback processes”.

8) Overall response from authors to my minor comments:

I find the responses to my individual "minor" queries a bit too basic. For example I had a whole list of minor comments and corrections:

Minor comments and corrections

(1) abstract, line 24: define "WPS"

(2) abstract line 24: the authors need to be much more precise here: it is change to surface temperature, and it is the meridional pattern, not the global value that is being discussed (compare with figure S8, where the explained variance of the surface pattern is much lower).

(3) Abstract line 26: see major comments above. The abstract talks of these feedbacks "driving uncertainties" where there needs to be much more discussion of causality

(4) abstract line 32: missing the word "only" before "... A little"

(5) line 49: need to insert the word "meridional" before "warming projection"

(6) line 74: delete 'an' before easy.

(7) line 99: dominate -> dominant

(8) line 119: isolations -> isolation

(9) line 123: insert 'regional' before WPS to contrast it with the global discussion

(10) line 125-128: please clarify how it is established that the spatial pattern or warming spread is caused by a "strong positive coupling" between these feedbacks

(11) define PWA in line 130

(12) line 138: as discussed above, why is the energy convergence change associated with surface turbulent heat fluxes a "feedback". Isn't it just a component of the surface energy budget?

(13) line 160: discussions -> discussion

(14) lines 165-7: see major comments above. Surely this should have the words 'regional patterns of' in front of 'climate change projections'

(15) line 406: scattering -> scatter

In answer to these 15 comments the authors simply respond: "We have made the suggested corrections/changes accordingly". As a reviewer it is extremely onerous to track down what was changed and how. Some of my comments were purely editorial, such as comments 4-9 where I don't need a specific response (although even here in acknowledgement that they made each particular change would be appreciated). However a number of my comments required changes to the way things were expressed by the authors, the detail provided, or asked for further explanation. This includes comments 2, 3, 10, 12, 14. Even if these comments are covered elsewhere in their response, I would expect specific responses to these comments.

Response: We appreciate your efforts in providing thorough reviews of our manuscript. We apologize not providing detailed information of our response to the minor comments (1) to (15) in the previous revision. Below are our point-to-point responses to the 15 minor comments that had been incorporated in our last round revision. All the line numbers and changes correspond to the information of the last round revision.

(1) "WPS" had been defined in line 25 as "*Warming Projection Spread*" (which has been rephrased as "*Warming Pattern Spread*" in the current revised version).

(2) In the resubmitted manuscript, we have pointed out that the global mean values of the EOF1 and EOF3 of the meridional pattern explained 98.7% of the global mean WPS in the main part. And we also noticed that the abstract in the resubmitted manuscript is still not precise enough. Therefore, we have made more precise statement in this round of revision as follows: "*This paper focuses on feedbacks that control the inter-model differences in their warming patterns or the "warming pattern spread" (WPS). WPS limits our ability to foresee the severity of regional impacts on nature and society. This study identifies two dominant patterns of WPS whose global mean values explain 98.7% of the global warming spread (GWS)...*"

(3) We changed the word "drives" to "*explains*" in the resubmitted manuscript. By doing this, this statement no longer necessarily implies a causal relation.

(4) This sentence had been changed to "*Other feedback spreads, including the cloud feedback spread, contribute less to the WPS as they tend to cancel each other out in a model-dependent manner.*". And "a little" had been changed to "*less*". (line 35)

(5) We added "*spatial*" before "warming projection" (line 53).

(6) "an" before "easy" had been deleted (line 77).

(7) The subtitle had been changed to "*Relationship between the global mean WPS and its meridional patterns*" (line 103). Now this has been further shortened as "*Relationship between the GWS and WPS*" following the newly defined abbreviations of GWS (global warming spread) and WPS (warming pattern spread) per your suggestions.

(8) "isolations" had been changed to "isolation" (line 124).

(9) "spatial pattern" had been added to contrast it with the global mean (line 121).

(10) "strong positive coupling" in the first version means that "models with a greater (smaller) albedo feedback also have a larger (smaller) water vapor feedback" as the EOF1 mode is mostly explained by these two feedbacks. These statement had been revised as follows: "*It follows that EOF1, the most dominant spatial pattern for the zonal and global mean WPS, is primarily caused jointly by ice-albedo and water vapor feedbacks such that models with a greater (smaller) albedo feedback also have a larger (smaller) water vapor feedback. Their collective effect leads to a stronger (weaker) warming at all latitudes with a more (less) pronounced PWA. The dominant EOF1 mode and the positive correlation of the ice-albedo feedback in high latitudes*

and water vapor feedback in low latitudes can also be inferred from a regression analysis against the global mean WPS (Figs. S4-S5)”

(11) “PWA” had been defined (line 34).

(12) “feedback” had been deleted in the revised manuscript (line 146).

(13) “discussions” had been changed to “*discussion*” (line 242).

(14) We believe that the newly added TOA-based analysis does further substantiate the statement: “This result brings into question as to whether cloud feedback is the leading uncertainty in climate change projections, particularly their spatial patterns”. Indeed, both the TOA-based (Fig. 6) and surface-based (Fig. 4) analyses reveal that only about 35% of the inter-model spread of the cloud feedback is projected onto the leading three meridional patterns of the inter-model WPS, which explains nearly 99% of the global mean WPS (Fig. 2).

(15) “scattering” had been changed to “scatter” (line 514).

9) Figures 4, 5, 6: all need labels on the vertical axis

Response: Thanks. We have added labels for vertical axis in Figs. 4, 5, and 6.

10) figure 5: please describe in the caption the criteria that was chosen for the ordering of the models

Response: Thanks. We add one sentence in figure caption to describe the criteria. “*Models are in the order of their projected global mean surface warming from the smallest to the largest.*”

11) figure S2: I don't understand this. How can the correlation between zonal mean temperature changes (i.e. a function of latitude) and the global mean temperature change (a single number) give a single value? Shouldn't this figure be a latitudinal plot?

Response: The correlation is done between the GWS (which is the global mean) and the global means of these zonal mean partial temperature changes. To make it clear, we have revised the figure caption as

Figure S2. Process Correlations with the GWS. Variance of zonal mean inter-model spreads of partial surface temperature changes due to individual feedback processes (blue, the ordinate on the left) and the correlations of their global means (orange, the ordinate on the right) with the global mean warming spread (GWS). The labels on the bottom of each panel stand surface albedo (AL), water vapor (WV), clouds (CLD), atmospheric dynamics (ATM), ocean dynamics/storage (OCN), and surface fluxes (HF) feedbacks.

Reviewers' comments, second round:

Reviewer #1 (Remarks to the Author):

This is a solid study and ready for publication. The authors considered the reviews seriously and revised the manuscript with additional analyses presented in response. The three reviews were very similar and I feel that the authors did a good job at answering concerns regarding the role and relationship of cloud feedbacks in the study and the surface-TOA differences. The manuscript as it is now provides the reader with adequate information to interpret and further explore the implications for climate models.

One typo on L 183 - 'differently' should be 'different'

Reviewer #2 (Remarks to the Author):

I see that the Editor has decided to go forward with potential publication in this non-specialized journal despite my advice of the contrary. I think the study in general has merit, even if it uses the out-dated climate model ensemble. The authors have done well in addressing my key comments.

Below I now only list a few specific remarks that I recommend should be addressed before publication.

154-56: The general idea is that regional temperature changes approximately scale with the global-mean warming. I don't think the authors invalidate this hypothesis. As such, the first sentence needs to be revised, more clear, or dropped. The second sentence in fact implies Arctic warming scales with the global mean one, which implies the former sentence (154 – 155) is wrong.

168-70: This is an obscure use of the term "feedback". A feedback in general is defined as a process that changes as a function of the global-mean surface temperature, and itself impacts the top-of-atmosphere radiation balance. I would guess a non-radiative feedback is a biogeochemical feedback (carbon cycle), but the keywords "ocean", "atmospheric dynamics" or "surface turbulent heat fluxes" do not resonate with any pre-described feedback process. A very careful rewording is necessary that explains exactly what the authors have in mind.

177: What do the authors mean by "magnitude of the ... pattern"? is it the amplitude, i.e. difference between least and most warming regions, or is it the magnitude, i.e. the overall amount of warming?

178-81: Fig. 1 suggests the authors think of the WPS as zonal, temporal mean meridional distribution of the warming. If so, this would be good to clarify at this point. If not, a clarification would help even more.

Reviewer #3 (Remarks to the Author):

Re-review of A Less Cloudy Picture of the Inter-Model Spread in Future Global Warming Projections, by X. Hu et al.

Many/most of my queries have been answered by the authors. I think that the results that they show are interesting and quite important ones in that they shed light on the causes of different warming pattern changes in GCMs. This is an important new perspective on the role of feedbacks and climate change. They also shed light on some differences between surface and top of atmosphere changes and feedbacks. The extension of the analysis to the top of atmosphere is a very nice addition. However, I still find the way that they present the results confusing and think that upfront clarification and then wording changes throughout would clarify their presentation of results, make it much easier for the reader to understand and make their points of difference from

earlier studies easier to understand.

1) My understanding of the paper is as follows: The key point of the paper and the way it differs from previous studies on the importance of feedbacks in model projection spread is twofold. Firstly the present study considers a surface perspective for evaluating feedbacks in models rather than the conventional top of atmosphere approach. Secondly, it focuses primarily on the impact of different feedbacks on the pattern of surface temperature change, rather than on the spread in global mean warming.

This needs to be stated very clearly upfront. It could be better to define WPS as "warming pattern spread", rather than "warming projection spread" as this makes it clearer. Secondly when global temperature changes are being considered I strongly suggest you refer to them as "global temperature spread (GTS)", to differentiate the pattern from the global values discussion.

2) Abstract:

Therefore, I suggest in the abstract that the authors insert texts such as.

Both the global magnitude of the warming, and the geographical pattern of that warming however differ substantially between models. This paper focuses on feedbacks that control the pattern of warming. We refer to differences across models in their pattern of warming as "warming projection (or pattern) spread" (WPS). WPS limits our ability to foresee the severity of regional impacts on nature and society.

Further down in the abstract I don't understand what is meant by two patterns of warming explaining 97% of global temperature spread. Don't you mean 90% of the meridional pattern spread?

3) Introduction:

the first paragraph is talking about purely global temperature spread. Therefore drop all references in this paragraph to WPS, and change it to "global warming spread (GWS)" or something similar. There is no reference to patterns here therefore use of WPS is confusing.

4) Throughout the paper, whenever WPS refers to global changes, e.g. "global mean WPS", I suggest you change it to GWS. So for example, in the first line of the second paragraph change WPS to GWS.

5) Further down there seems to be a second definition of WPS: the text says "We define the inter-model WPS as the range in model simulated global mean warming after removing the MME average (Fig. 1c). For easy reference, we call the departures of individual models' temperature changes from the MME as the WPS."

This is very confusing: why is the WPS now defined as global mean warming, rather than the pattern? Incidentally figure 1C shows meridional changes, not global changes.

6) It is pleasing that the authors have now performed further calculations normalising the radiative changes by the global temperature change (lines 206-231) I have a number of comments about this specifically:

a. firstly, it deserves its own section. It sits in the section "top of atmosphere versus surface" but it is not about that surface/TOA issue. Rather it is about the issue of expressing changes as feedbacks, rather than absolute flux changes.

b. Secondly, the authors need to make explicit why normalising by surface temperature changes suddenly brings their results in line with results elsewhere and e.g. supported in the last 3 IPCC reports. It is not enough to just say "normalising by surface temperature make this happen" they need to say what is the essential difference now in the analysis when normalised by surface temperature that now makes the methodologies agree. I suspect it is because normalisation expresses the CFRAM flux changes instead as true feedbacks (W/m²/K) rather than simply as flux changes (W/m²). It could be argued the former are the cause of the global temperature spread, the latter are the result of the global spread. The authors need to discuss this.

7) Section starting line 181:

Related to the point above: when discussing the discrepancy with other results, one important issue is that the present study is looking at absolute changes in the surface fluxes which correspond with the spread in temperature between models. Although the authors use the word "feedbacks" to describe processes like surface albedo and water vapour responses they are not really feedbacks in the way that it is defined elsewhere in the literature, where radiative changes

are normalised by global surface temperature changes ($W/m^2/K$). Therefore the authors need to specifically clarify in the text up front somewhere that where they refer to "feedbacks", this is shorthand for the net flux changes due to particular processes.

8) Overall response from authors to my minor comments:

I find the responses to my individual "minor" queries a bit too basic. For example I had a whole list of minor comments and corrections:

Minor comments and corrections

(1) abstract, line 24: define "WPS"

(2) abstract line 24: the authors need to be much more precise here: it is change to surface temperature, and it is the meridional pattern, not the global value that is being discussed (compare with figure S8, where the explained variance of the surface pattern is much lower).

(3) Abstract line 26: see major comments above. The abstract talks of these feedbacks "driving uncertainties" where there needs to be much more discussion of causality

(4) abstract line 32: missing the word "only" before "... A little"

(5) line 49: need to insert the word "meridional" before "warming projection"

(6) line 74: delete 'an' before easy.

(7) line 99: dominate -> dominant

(8) line 119: isolations -> isolation

(9) line 123: insert 'regional' before WPS to contrast it with the global discussion

(10) line 125-128: please clarify how it is established that the spatial pattern or warming spread is caused by a "strong positive coupling" between these feedbacks

(11) define PWA in line 130

(12) line 138: as discussed above, why is the energy convergence change associated with surface turbulent heat fluxes a

"feedback". Isn't it just a component of the surface energy budget?

(13) line 160: discussions -> discussion

(14) lines 165-7: see major comments above. Surely this should have the words 'regional patterns of' in front of 'climate change projections'

(15) line 406: scattering -> scatter

In answer to these 15 comments the authors simply respond: "We have made the suggested corrections/changes accordingly". As a reviewer it is extremely onerous to track down what was changed and how. Some of my comments were purely editorial, such as comments 4-9 where I don't need a specific response (although even here in acknowledgement that they made each particular change would be appreciated). However a number of my comments required changes to the way things were expressed by the authors, the detail provided, or asked for further explanation. This includes comments 2, 3, 10, 12, 14. Even if these comments are covered elsewhere in their response, I would expect specific responses to these comments.

9) Figures 4, 5, 6: all need labels on the vertical axis

10) figure 5: please describe in the caption the criteria that was chosen for the ordering of the models

11) figure S2: I don't understand this. How can the correlation between zonal mean temperature changes (i.e. a function of latitude) and the global mean temperature change (a single number) give a single value? Shouldn't this figure be a latitudinal plot?

Response to Reviewer 3:

Reviewer #3 (Remarks to the Author):

Review of " A Less Cloudy Picture of the Inter-Model Spread in Future Global Warming Projections " by Hu et al. [Manuscript Number: NCOMMS-19-42302].

This paper investigates the spread in surface temperature response from CMIP5 models under RCP 8.5, and uses the "CFRAM" methodology to isolate surface temperature change contributions from individual processes including water vapour, clouds, turbulent fluxes, oceanic convergence, surface albedo et cetera. The paper uses an EOF approach to define the major meridional patterns of temperature change, then correlates these with the spreads in the surface temperature change components. It concludes that water vapour is the dominant feedback causing change at low latitudes, with surface albedo dominant at high. It concludes that reductions in uncertainties in these two feedbacks should be a priority for reducing spread in global temperature projections.

The authors are to be commended for exploring the surface contributions to temperature change, and relating these to individual processes and feedbacks, with the aim of understanding the causes of the meridional pattern changes. The CFRAM methodology is suited to this, and the analysis for the most part is a sound application of this. I do have some significant issues, however with the conclusions reached, particularly with regard to the role and importance of the individual feedbacks, and the consistency or otherwise with findings elsewhere that take a top of atmosphere view. There are also a range of clarifications needed. These issues would need to be addressed before the paper was suitable for publication.

Response: We thank the reviewer for the compliments and the succinct summary of our work. In the revision, we have fully taken all of your valuable and constructive comments/suggestions into consideration. In particular, we have added a new section ("**Comparison with the TOA perspective**") and four new figures (Fig.6, and Figs. S7-S9), as well as amending Fig. 4 (which was Fig. S7) with two new panels: Figs. 4c and 4d, to address the main comment raised by all reviewers, concerning if the difference between ours and previous studies is due to (i) the surface versus TOA perspectives and (ii) the consideration of spatial patterns. The main results of the new TOA-based analysis are: (1) Both the TOA-based and surface-based analyses reveal the importance of the cloud feedback for the global mean WPS (Figs. 4c-4d and S2 for the surface, Figs. 6c-6d and Fig. 9a for the TOA); (2) When the partial radiative flux perturbations at the TOA for each model is normalized by their respective global mean surface temperature change, **we can reproduce the results found in previous studies, namely, the inter-model spread of the cloud feedback parameter can be singled out as the main contributor, among all feedback parameters, to the global mean WPS** (Fig. S9b); (3) We also confirm that when not normalizing by the total global mean temperature change, both perspectives reveal nearly equal important roles of the water vapor, clouds, and ice-albedo feedbacks for the global mean WPS (Fig. S2 for the surface and Fig. S9a for the TOA); (4) The regression analysis of the TOA-based feedbacks against the leading three EOF modes confirm our conclusions made from the surface perspective, including that only fractional inter-model spread of the cloud feedback (about 35%) is projected to the three EOF modes (Figs. 3-4 for the surface and Figs. S8 and 6 for the TOA). Therefore, we conclude that "*the difference between the TOA and surface perspectives does not account for the*

difference of our results with previous studies”; and “there is no real discrepancy between our results and previous studies in terms of the global mean WPS because it can be easily resolved when the PRPs at the TOA are normalized by the global mean surface temperature change. The consideration of the inter-model spread in the meridional surface warming pattern reduces the cloud feedback’s contribution to the WPS, relative to the water vapor and ice-albedo feedbacks”.

In the revision, we found that one of the 26 models (FGOALS-s2) considered in the previous submission has the strongest negative cloud feedback in the tropics among all models despite it produces the strongest global mean warming (see Fig. A1 in the end of the response letter). As a result, the inclusion of this model would degrade the correlation between the WPS and inter-model spread of the cloud feedback from both the TOA and surface perspectives and with and without considering the spatial patterns. In particular, the inclusion of this model would wipe out, almost entirely, the strong positive correlation between the global mean WPS and the inter-model spread of the cloud feedback parameter, although it does not change the characteristics of other results in any meaningful way. For this reason, we only present the results based on the 25 models’ climate simulations.

Major comments:

(1). A much clearer discussion of the contrast between surface and TOA feedbacks needs to be made. An extremely large number of papers and studies, along with the last three IPCC reports, conclude that cloud uncertainties dominate the uncertainty in the climate sensitivity and the WPS at a global scale. Furthermore, the processes controlling (top of atmosphere) water vapour/lapse rate feedbacks are relatively well understood, as is the anti-correlation between them, and the large uncertainty in cloud processes and their impact on the Earth’s atmospheric radiative balance (both rapid response and feedback) are clearly established. The conclusions of this paper sit at variance with this – or at least are expressed to be at variance with this, therefore the authors must establish clearly how their surface energy budget perspective provides a different interpretation. I suspect there is ultimately consistency between the two, because they are addressing different issues – top of atmosphere response of processes controlling global temperature response, and net energy imbalances controlling surface meridional warming patterns. The authors need to clearly articulate that these are different and what drives the differences.

One place to start could be to also perform a TOA analysis along the same lines and compare the results. My understanding is that CFRAM can produce such an analysis, and therefore the contrast could be made. Another issue to address is, since the lapse rate feedback is not included in CFRAM as a separate process, is this a major factor helping to drive the differences between surface/TOA? The authors also need to discuss these differences at length, so that the differences in TOA/surface interpretations can be understood.

Response: Following the reviewers’ suggestion, we have made the exact same analysis using the partial radiative flux perturbations at TOA. The results confirm your conjecture that “*there is ultimately consistency between the two*”. Specifically, as shown in the new figures (Figs. 6 and S7-S9) and summarized in the new section (“**Comparison with the TOA perspective**”), the TOA-based feedback analysis basically reveals the same results as the surface-based analysis. Only when we divide the partial radiative flux perturbations at TOA by the global mean surface temperature change (i.e, climate feedback parameters) does the TOA picture deviate from that given by the surface perspective for the radiative feedbacks shared by both

perspectives. The fact that our results derived from the feedback parameter are in agreement with all of previous studies imply that *“there is no real discrepancy between our results and previous studies in terms of the global mean WPS because it can be easily resolved when the PRPs at the TOA are normalized by the global mean surface temperature change”*. In the new section, we have also discussed the results of the lapse rate feedback.

(2). The above comment related to global differences in temperature response. At the meridional level there is also inconsistency between the current results and those of earlier studies (e.g. Vial et al, Bonan et al., 2018). For example, the Bonan et al states that "cloud feedbacks, in particular, contribute the greatest source of warming uncertainty in most regions". The present paper needs to address/explain its differing conclusions at the meridional and at the global level.

Response: As summarized above, we have explained the different conclusion at the global level in the revision. We only found one paper authored by Vial et al. in 2018 on her website, which is not about inter-model spread but focuses on ensemble experiments of a single model. Therefore, we have not included it in our reference. We already cited Bonan et al. (2018) in the original submission. After reading through their paper very carefully, we feel that the differences between their and our conclusions cannot be easily reconciled without running their “moist” energy balance model. Therefore, we are not in the position to comment on their results in our manuscript publicly.

Here we wish to share our interpretation on their results privately. In essence, what they did is to construct a 1D (meridional) “moist” energy balance model forced by anthropogenic radiative forcing and ocean heat uptake derived from CMIP5 climate simulations. They assumed the total longwave energy emission anomaly is linearly proportional to the temperature, as any one-layer energy balance model. They used the total feedback parameter derived from the CMIP5 climate simulations as the coefficient for the longwave wave emission. By using outputs derived from each individual CMIP5 simulation for the meridional profiles of (i) anthropogenic forcing, (ii) ocean heat uptake, and (iii) the total climate feedback parameter, they more or less reproduced the meridional profile of the surface warming of each individual CMIP5 simulation. By repeating the same calculation using outputs for all CMIP5 simulations under consideration, they more or less reproduced the inter-model spreads in the meridional pattern of surface warming. It should be pointed out the only thing that would prevent them to exactly reproduce the meridional pattern of the CMIP5 simulations is that they used a fixed coefficient for the diffusive term which represents the poleward energy transport in nature. In other words, it is not surprised that such good results are achieved since all terms but one (diffusive term) are derived from model outputs.

To evaluate the impact of the inter-model spread of an specific feedback (say cloud feedback), they subtracted the MME of the feedback parameter of that specific feedback from the MME of the total feedback parameter and then added the feedback

parameter of the same feedback derived from one individual CMIP5 model output and run the moist energy balance model. They repeated this using the feedback parameter of the same feedback derived from each CMIP5 model simulation. The difference of these experiments of their moist energy balance corresponds to the inter-model spread of the meridional pattern of the surface warming due to that specific feedback. By design, the feedback parameter that has largest inter-model spread would automatically produce the largest spread in their energy balance model experiments. In other words, we believe their results are made by design. In our study, we let the data of the original climate simulations naturally indicate what processes are responsible for the WPS.

(3) The authors should clarify the physical meaning of surface non-radiative "feedbacks". For example, how do changes in convective or advective convergence "feed back" on the surface temperature change? For example, does increased surface warming/cooling due to changes in turbulent fluxes in turn reinforce/oppose those flux changes and further increase surface warming (i.e. feed back' on it?? At the TOA the process is clear, but at the surface less so.

Response: To continue with the example given by the reviewer: Let's say the surface warms increasing the surface-to-air temperature gradient, which in turn increases the surface-to-air sensible heat flux. The contribution of the increase in sensible heat flux is to cool the surface and warm the atmosphere by increasing the transfer of energy from the surface to the atmosphere. This in turn once again modifies the surface-to-air temperature gradient, which again modifies the sensible heat flux transfer and in consequence its contribution to the surface temperature change. This feedback mechanism is not only dependent on surface temperature change but also on the lower atmospheric temperature change, so in this way its feedback mechanism is more complicated. To avoid any confusion, however, we no longer reference changes in surface turbulent fluxes as feedbacks explicitly.

(4) Earlier studies have found that because of nonlinearities in radiative response to clouds (overlap etc.), using long-term average for clouds do not provide an accurate representation of the radiative impact of cloud changes (e.g. early papers on the Partial Radiative Perturbation approach – Wetherald and Manabe, 1988, and a number of later papers). To overcome this, PRP typically uses daily cloud fractions in radiation calculations. Yet the approach here uses 50 year mean clouds in its calculations (e.g. lines 279+). Considering critical conclusions are being made about the strength and spread of the cloud feedback, the authors need to establish that cloud feedbacks are accurately represented from CFRAM using long-term means.

Response: As far as we know, the majority of climate feedback studies use the longtime average fields to estimate partial radiative flux perturbations (PRP) that are exclusively due to changes in clouds fields. Let us use Zelinka et al. (2018) as an example, who show a high correlation between inter-model spreads of the global mean surface temperature change and the cloud feedback parameter under the scenario of equilibrium climate response to an abrupt quadratic CO₂ forcing (0.62 for CMIP5 models and 0.81 for CMIP6 models, in their Fig. S4). According to their description, they estimate PRP due to the cloud feedback by “multiplying the (seasonally and) spatially-resolved kernels with the relevant (monthly) climate field

anomalies” and “then annually averaged to produce a 150-year time series of TOA radiative flux anomalies due to each field”. Because (i) the feedback kernels are pre-calculated using the climatological fields of the control climate statement and (ii) the product of monthly anomaly fields with kernels is a linear calculation, the 150-year average of the monthly products would be the same as the product of the longtime average anomaly field with kernel if we ignore the annual cycle of the kernels. Therefore, what Zelinka et al. (2018) did is equivalent to our approach using longtime mean clouds fields other than they calculated cloud feedbacks by resolving climatological annual cycle. As shown in our new Fig. 8b, we have more or less reproduced Zelinka et al. (2018)’s results, showing the correlation of the WPS and the inter-model spread of the cloud feedback parameter of these 25 CMIP5 RCP8.5 simulations is about 0.5. The slightly smaller correlation is mainly due to the transient climate response versus equilibrium climate response because the later typically have a larger correlation. In that regard, we believe that our cloud feedback calculations are as accurate as the existing studies.

(5). The authors need to clarify that it is actually spread in water vapour "feedback" that causes the spread in surface temperature change. I mean here the radiative response from water vapour changes per degree of surface warming, not just the radiative changes resulting from water vapour changes. A surface temperature increase (particularly in the tropics), will always be associated with warmer, moister air at low levels which will more strongly radiate to the surface. So a warmer surface will be necessarily associated with stronger downwelling longwave radiation from water vapour, but it doesn't mean that water vapour is causing that extra warming (it may simply be a result of the warming). To paraphrase: is the water vapour term normalised by the surface temperature change, or is it just the absolute change in downwelling radiation from water vapour changes?

Response: We don't normalize the water vapor feedback term, nor any feedback terms.

The reviewer is correct that there is a strong coupling between greater surface warming and greater downwelling LW radiation from water vapor among models. It is precisely this strong consistent coupling throughout the model ensemble that makes the water vapor feedback a leading contributor to the WPS. The water vapor feedback, is precisely termed a feedback, because surface warming leads to an increase of water vapor that leads to more downwelling LW radiation to the surface that leads to more surface warming, so on and so forth. Just because the surface warming causes more water vapor does not mean the greater water vapor is not the cause of more surface warming. Granted there are other processes responsible for the surface warming and also the atmospheric warming (that controls the saturation vapor pressure), but that is precisely what the CFRAM and any other partial radiative perturbation methods seeks to separate is the contribution to the surface warming by that process alone. So the warming contribution given by CFRAM for the water vapor feedback is not the total surface warming but that due to the water vapor feedback alone.

The CFRAM-based surface feedback analysis is built on the energy balance equation. Unlike the TOA-based feedback analysis in which the feedbacks are addible (i.e., the

total feedback parameter is equal to the sum of individual feedback parameters) but their effects are not addible (i.e., one cannot determine partial temperature changes due to individual feedbacks). In CFRAM-based feedback analysis, feedbacks (i.e., partial energy flux perturbations) are addible and so are their effects (i.e., partial temperatures) are addible. The sum of these partial temperature changes calculated by CFRAM can be directly compared to the total temperature changes derived from observations or original model simulations. In other words, CFRAM calculates partial temperature changes and their sum without using the information of the total temperature changes. The partial surface temperature change due to a feedback (e.g., the water vapor feedback) has two parts: one is directly related to the “absolute change in down welling radiation” due to change in the feedback agent (e.g., water vapor changes) and the other part is related to change in the down welling radiation due to air temperature change in response to the feedback agent induced radiative flux convergence perturbation in the air (the second part corresponds to the lapse rate feedback associated with the feedback agent). Therefore, the change of the surface temperature is purely due to changes in that specific feedback agent.

Lastly, we wish to point out that the normalization by the total surface temperature change in effect removes the relation of feedbacks to the inter-model WPS. For example, if we were trying to de-seasonalize a monthly data set, we would divide the monthly data set by its respective climatological monthly mean and then add the resulting value to the climatological annual mean to obtain a de-seasonalized data set. This would allow for an apples-to-apples comparison of the data set since this would remove the influence of the seasonal variance. Similarly, dividing by the respective surface temperature change of each model removes the influence of the inter-model variance in the surface temperature change.

(6) The paper is pretty vague about "positive coupling" between e.g. water vapour and surface albedo feedbacks. Please define precisely what this means, and what the evidence is for it.

Response: The phrase “positive coupling” merely refers to a positive correlation between them. In the revision, we have replaced the phrase “positive coupling” with “positive correlation”.

(7) The paper discusses the range in the global WPS, and the meridional pattern of WPS. To aid the reader, the authors should make it clearer which they are talking about and when. Jumping between the two is confusing. E.g. in lines 104 to 118. I suggest the paper has different sections on global mean WPS spread and the spread in the meridional patterns (i.e. regional WPS).

Response: This work focuses on the relationship between the global mean WPS and its meridional pattern. To make it clear, we have changed the section subtitle from “**Dominant spatial patterns of WPS and their contributions to the global mean WPS**” to “**Relationship between the global mean WPS and its meridional patterns**”

(8) in summary a number of significant issues need to be addressed before the claim made in the last sentence of the abstract, or in lines 165-167 can be substantiated. I encourage the authors to address these issues, as their approach is an interesting and worthwhile one and a surface perspective on feedbacks and feedback spread would be extremely useful in the literature.

Response: We believe that the newly added TOA-based analysis does further substantiate the statement: “This result brings into question as to whether cloud feedback is the leading uncertainty in climate change projections, particularly their spatial patterns”. Indeed, both the TOA-based (Fig. 6) and surface-based (Fig. 4) analyses reveal that only about 35% of the inter-model spread of the cloud feedback is projected onto the leading three meridional patterns of the inter-model WPS, which explains nearly 99% of the global mean WPS (Fig. 2). Fig. S2 (surface) and Fig. S9a (TOA) also indicate that when not normalizing by the total global mean temperature change, both perspectives reveal nearly equal important roles of the water vapor, clouds, and ice-albedo feedbacks for the global mean WPS.

(9) I found a number of figures hard to digest. These include figure 4 and supplementary figures 3, 4 and 5. It would help the reader if these models were not defined in numerical order (i.e. alphabetical order) but something more meaningful to the discussion -- strongest to weakest temperature response, or some other relevant physical parameter.

Response: We are very grateful for this suggestion, which makes the results vividly clear (see the revised Fig. 5).

Minor comments and corrections

(1) abstract, line 24: define "WPS"

(2) abstract line 24: the authors need to be much more precise here: it is change to surface temperature, and it is the meridional pattern, not the global value that is being discussed (compare with figure S8, where the explained variance of the surface pattern is much lower).

(3) Abstract line 26: see major comments above. The abstract talks of these feedbacks "driving uncertainties" where there needs to be much more discussion of causality

(4) abstract line 32: missing the word "only" before "... A little"

(5) line 49: need to insert the word "meridional" before "warming projection"

(6) line 74: delete 'an' before easy.

(7) line 99: dominate -> dominant

(8) line 119: isolations -> isolation

(9) line 123: insert 'regional' before WPS to contrast it with the global discussion

(10) line 125-128: please clarify how it is established that the spatial pattern or warming spread is caused by a "strong positive coupling" between these feedbacks

(11) define PWA in line 130

(12) line 138: as discussed above, why is the energy convergence change associated with surface turbulent heat fluxes a "feedback". Isn't it just a component of the surface energy budget?

(13) line 160: discussions -> discussion

(14) lines 165-7: see major comments above. Surely this should have the words 'regional patterns of' in front of 'climate change projections'

(15) line 406: scattering -> scatter

Response: We have made the suggested corrections/changes accordingly.

(16) supplementary table 3: what does "P" mean under the vertical resolution column? (Also what does S stand for? Presumably surface?)

Response: We have provided the information for the letters “P” and “S” in Table S3.

(17) Please clarify more carefully the purpose of supplementary figures 3 and 4.

Response: As stated in the main text, Fig. S3 displays the principal components of the first three EOF modes, which are used for the regression analysis of individual feedback terms. To make it clear, we have referenced this figure when first mentioning regression analysis.

In the revision, we have removed the original Fig. S4.

(18) I found the caption and figure S5 and figure S6 hard to understand.

Response: The captions have been revised.

Figure A1. Partial Radiative Energy Flux Perturbations at the TOA. Zonal means of partial radiative energy flux perturbations (PRP, W/m^2) given by all 26 models. Panels (a)–(f) are PRP associated with, respectively, the external forcing (EXT), Planck feedback (PL), surface albedo feedback (AL), water vapor feedback (WV),

cloud feedback (CLD), and lapse rate feedback (LR). The solid line in each panel is the MME mean and the dashed thick line corresponds to the FGOALS-s2 model's results.

Figure A2. Partial Surface Temperature Changes. Zonal means of CFRAM-derived partial surface temperature changes given by all 25 models. Panels (a)–(h) are CFRAM-decomposed partial surface temperature changes due to changes in, respectively, the external forcing (EXT), surface albedo (AL), water vapor (WV), clouds (CLD), atmospheric dynamics (ATM), ocean dynamics and heat storage (OCN), surface heat fluxes (HF), and ozone (O3). Panel (i) is the sum of (a)–(h) and (j) is the difference between Fig. 1b and the panel (i), corresponding to the error of the CFRAM analysis. The solid line in each panel is the MME mean and the dashed thick line corresponds to the FGOALS-s2 model’s results.

Response to Reviewer 2:

Reviewer #2 (Remarks to the Author):

Hu et al. analyse model output of the former phase of the Coupled Model Intercomparison Project (CMIP5) to assess the pattern of warming in 2100 in the RCP8.5 scenario in relationship to feedbacks quantified at the surface. This study differs from previous analyses in two aspects: (1) feedbacks are defined at the surface, and include thus additional processes, and (2) the authors focus on the geographical distribution of warming, rather than on its global amount.

Response: We thank the reviewer for taking time in reviewing our manuscript. In the revision, we have fully taken all of your valuable and constructive comments/critiques into consideration. In particular, we have added a new section (“**Comparison with the TOA perspective**”) and four new figures (Fig.6, and Figs. S7-S9), as well as amending Fig. 4 (which was Fig. S7) with two new panels: Figs. 4c and 4d, to address the main comment raised by all reviewers, concerning if the difference between ours and previous studies is due to (i) the surface versus TOA perspectives and (ii) the consideration of spatial patterns. The main results of the new TOA-based analysis are: (1) Both the TOA-based and surface-based analyses reveal the importance of the cloud feedback for the global mean WPS (Figs. 4c-4d and S2 for the surface, Figs. 6c-6d and Fig. 9a for the TOA); (2) When the partial radiative flux perturbations at the TOA for each model is normalized by their respective global mean surface temperature change, **we can reproduce the results found in previous studies, namely, the inter-model spread of the cloud feedback parameter can be singled out as the main contributor, among all feedback parameters, to the global mean WPS** (Fig. S9b); (3) We also confirm that when not normalizing by the total global mean temperature change, both perspectives reveal nearly equal important roles of the water vapor, clouds, and ice-albedo feedbacks for the global mean WPS (Fig. S2 for the surface and Fig. S9a for the TOA); (4) The regression analysis of the TOA-based feedbacks against the leading three EOF modes confirm our conclusions made from the surface perspective, including that only fractional inter-model spread of the cloud feedback (about 35%) is projected to the three EOF modes (Figs. 3-4 for the surface and Figs. S8 and 6 for the TOA). Therefore, we conclude that “*the difference between the TOA and surface perspectives does not account for the difference of our results with previous studies*”; and “*there is no real discrepancy between our results and previous studies in terms of the global mean WPS because it can be easily resolved when the PRPs at the TOA are normalized by the global mean surface temperature change. The consideration of the inter-model spread in the meridional surface warming pattern reduces the cloud feedback’s contribution to the WPS, relative to the water vapor and ice-albedo feedbacks*”.

In the revision, we found that one of the 26 models (FGOALS-s2) considered in the previous submission has the strongest negative cloud feedback in the tropics among all models despite it produces the strongest global mean warming (see Figs. A1-A2 in the end of the response letter). As a result, the inclusion of this model would degrade the correlation between the WPS and inter-model spread of the cloud feedback from

both the TOA and surface perspectives and with and without considering the spatial patterns. In particular, the inclusion of this model would wipe out, almost entirely, the strong positive correlation between the global mean WPS and the inter-model spread of the cloud feedback parameter, although it does not change the characteristics of other results in any meaningful way. For this reason, we only present the results based on the 25 models' climate simulations.

The authors analyse CMIP5 models that have been studied intensely before, the results in the literature assessing top-of-atmosphere, global quantities yield very consistent results that are adequately reported in the introduction: importance of the cloud feedback for the global temperature uncertainty, importance of surface albedo and lapse-rate feedback for Arctic amplification. In title and discussions the present manuscript seems to suggest that the previous literature is called into question by the results presented here. However, little evidence is provided. Rather, it seems very plausible that the difference in conclusion is due to the difference in question asked. One is the surface perspective: this is a very different approach compared to the top of the atmosphere (TOA). It is the energy budget at TOA that determines climate. Some feedbacks act similarly at surface and TOA, most show distinct differences in their effects. The decomposition is much less straightforward at the surface, where also the fluxes from below (land or ocean surface) need to be taken into account. This makes the interpretation more difficult.

Response: In the revision, we have further substantiated the evidence that fully supports our conclusion. Specifically, we have made the exact same analysis using the partial radiative flux perturbations at TOA. As shown in the new figures (Figs. 6 and S7-S9) and summarized in the new section (“**Comparison with the TOA perspective**”), the TOA-based feedback analysis basically reveals the same results as the surface-based analysis. Only when we divide the partial radiative flux perturbations at TOA by the global mean surface temperature change (i.e, climate feedback parameters) does the TOA picture deviate from that given by the surface perspective for the radiative feedbacks shared by both perspectives. The fact that our results derived from the feedback parameter are in agreement with all of previous studies imply that *“there is no real discrepancy between our results and previous studies in terms of the global mean WPS because it can be easily resolved when the PRPs at the TOA are normalized by the global mean surface temperature change”*.

The other aspect is the focus on pattern rather than overall magnitude. The cloud feedback has a noisy and not very distinct spatial pattern. The surface albedo effect has a very distinct pattern (strong and positive in high latitudes, zero in low latitudes) as has the water vapour feedback (exponentially related to surface temperature). It is thus not surprising that these two correlate well with distinct warming patterns.

Overall, the study is well conducted and instructive. I question, however, the way the results are presented, which seemingly construct discrepancies with previous literature due to the aim to publish in a non-specialized journal. So I would urge the authors to re-formulate their findings in a more balanced way, in which case the study may very well be worth publication, but better so in a specialized journal.

Response: We focus on both pattern and overall magnitude of inter-model spreads of surface warming and climate feedbacks. The former is represented by the latitudinal profiles whereas the latter is by the total variance as well as the amount of the variance that is projected on the dominant patterns. We use the difference between the

total variance and the projected variance to separate those processes that have a great contribution to the WPS from those that have less contribution (i.e., the smaller/larger difference is, the stronger/weaker contribution is). We believe this is what makes our study novel, enabling a less cloudy picture of the WPS. In the revision, we have confirmed the findings under the surface perspective are the same under a TOA perspective. We believe that the reconciliation and convergence of the two perspectives does provide a balanced explanation with full evidence.

Zonal-mean PRP at TOA ($W \cdot m^{-2}$)

Figure A1. Partial Radiative Energy Flux Perturbations at the TOA. Zonal means of partial radiative energy flux perturbations (PRP, W/m^2) given by all 26 models. Panels (a)–(f) are PRP associated with, respectively, the external forcing (EXT), Planck feedback (PL), surface albedo feedback (AL), water vapor feedback (WV), cloud feedback (CLD), and lapse rate feedback (LR). The solid line in each panel is the MME mean and the dashed thick line corresponds to the FGOALS-s2 model’s results.

Figure A2. Partial Surface Temperature Changes. Zonal means of CFRAM-derived partial surface temperature changes given by all 25 models. Panels (a)–(h) are CFRAM-decomposed partial surface temperature changes due to changes in, respectively, the external forcing (EXT), surface albedo (AL), water vapor (WV), clouds (CLD), atmospheric dynamics (ATM), ocean dynamics and heat storage (OCN), surface heat fluxes (HF), and ozone (O3). Panel (i) is the sum of (a)–(h) and (j) is the difference between Fig. 1b and the panel (i), corresponding to the error of the CFRAM analysis. The solid line in each panel is the MME mean and the dashed thick line corresponds to the FGOALS-s2 model’s results.

Response to Reviewer 1:

Reviewer #1 (Remarks to the Author):

This paper makes a nice case that the coupling among feedback processes in the climate system is important to better understand and represent accurately in models to decrease uncertainty in warming projections. The study uses statistical analyses (EOFs) of CMIP5 results to show that particular feedbacks are strongly coupled across the range of models. The results differ from previous studies owing to their use of geographical variations in the warming projection spread and feedback strengths as well as their use of a warming projection at the surface which provides insights that are not evident from global mean analyses done for top of the atmosphere changes in radiative fluxes.

Ultimately, the results show that a strong relationship between the ice-albedo feedback and water vapor feedback in the models leads to global patterns that are dependent on the strength of these two feedbacks and their relationship to each other. These feedbacks are also negatively related to the surface heat fluxes. The spread in the warming among the models is largely controlled by the strength of this group of feedbacks. While the absolute spread in the cloud feedback is large and has been identified as a major contributor to the spread in global mean warming projects, it is not systematically related to other processes in the model so may damp or enhance other feedbacks in space and time and thus is not the dominant contributor to the warming projection spread as it has been analyzed here.

The analysis is sound and well presented with all contributing data and methods clearly presented. The figures are clear. The analysis is complex so there are many figures but I think the authors have made the appropriate decisions about what to put in the main section of the paper versus the supplementary material.

I would recommend that this paper be published with attention to the editorial revisions listed below outside of one overarching concern.

Response: We thank the reviewer for the compliments and the succinct summary of our work. In the revision, we have fully taken all of your valuable comments/suggestions into consideration. In particular, we have added a new section (“**Comparison with the TOA perspective**”) and four new figures (Fig.6, and Figs. S7-S9), as well as amending Fig. 4 (which was Fig. S7) with two new panels: Figs. 4c and 4d, to address the main comment raised by all reviewers, concerning if the difference between ours and previous studies is due to (i) the surface versus TOA perspectives and (ii) the consideration of spatial patterns. The main results of the new TOA-based analysis are: (1) Both the TOA-based and surface-based analyses reveal the importance of the cloud feedback for the global mean WPS (Figs. 4c-4d and S2 for the surface, Figs. 6c-6d and Fig. 9a for the TOA); (2) When the partial radiative flux perturbations at the TOA for each model is normalized by their respective global mean surface temperature change, **we can reproduce the results found in previous studies, namely, the inter-model spread of the cloud feedback parameter can be singled out as the main contributor, among all feedback parameters, to the global mean WPS** (Fig. S9b); (3) We also confirm that when not normalizing by the total global mean temperature change, both perspectives reveal nearly equal important roles of the water vapor, clouds, and ice-albedo feedbacks for the global mean WPS (Fig. S2 for the surface and Fig. S9a for the TOA); (4) The regression analysis of the TOA-based feedbacks against the leading three EOF modes confirm our conclusions made from the surface perspective, including that only fractional inter-model spread of the cloud feedback (about 35%) is projected to the three EOF modes (Figs. 3-4 for

the surface and Figs. S8 and 6 for the TOA). Therefore, we conclude that “*the difference between the TOA and surface perspectives does not account for the difference of our results with previous studies*”; and “*there is no real discrepancy between our results and previous studies in terms of the global mean WPS because it can be easily resolved when the PRPs at the TOA are normalized by the global mean surface temperature change. The consideration of the inter-model spread in the meridional surface warming pattern reduces the cloud feedback’s contribution to the WPS, relative to the water vapor and ice-albedo feedbacks*”.

In the revision, we found that one of the 26 models (FGOALS-s2) considered in the previous submission has the strongest negative cloud feedback in the tropics among all models despite it produces the strongest global mean warming (see Figs. A1-A2 in the end of the response letter). As a result, the inclusion of this model would degrade the correlation between the WPS and inter-model spread of the cloud feedback from both the TOA and surface perspectives and with and without considering the spatial patterns. In particular, the inclusion of this model would wipe out, almost entirely, the strong positive correlation between the global mean WPS and the inter-model spread of the cloud feedback parameter, although it does not change the characteristics of other results in any meaningful way. For this reason, we only present the results based on the 25 models’ climate simulations.

In the discussion section the results are presented such that the largest contributors to WPS and largest uncertainties in climate projects are the same thing. Stated another way, could the uncertainty in the coupling of the cloud feedback with the other feedbacks change the fundamental relationship of this analysis enough that the spread in the cloud feedback is then the primary contributor to uncertainty?

Response: We agree with the point the reviewer makes. The uncertainties in the coupling of the cloud feedback with other feedbacks could impact the spreads of the other feedbacks and the WPS. However, until these coupling issues among the models is resolved, we do not know the extent of the impact uncertainties between process couplings have on the WPS. In the paper, we make clear that a resolution of the uncertainties in the coupling of processes among models is critical to resolving the WPS.

The message that this study presents – that reducing spread in model representations of ice-albedo feedback, water vapor feedback, and the relationship between the two will necessarily reduce model warming projecting uncertainty – is important. But the fact remains that there is large spread in the cloud feedback representations AND their relationships to other feedbacks and as currently represented there are dampening factors that reduce their contribution to the WPS analyzed in this study. A recent number of papers analyzing early outcomes from CMIP6 model runs (e.g., Gettelman et al 2019 GRL, Wyser et al 2019 GMD, Zelinka et al 2020 GRL) indicate that changes in model physics that affect cloud feedbacks are the largest contributors to increases in equilibrium climate sensitivity (ECS) and thus the primary factor in changing projections of warming. The impact on ECS and WPS may be two different ways to assess warming projects that provide different estimates on uncertainty. I would like to hear the author’s defense of their discussions and conclusions.

Response: We agree that changes in model physics that affect cloud feedbacks could most

definitely impact the cloud feedback contribution to the ECS and its inter-model spread. In other words, the fact the cloud feedback spread remains larger than other feedbacks by itself implies it indirectly contribute to the inter-model warming spread. If this is the case in CMIP6 models, then the cloud feedback could explain an increase of the WPS for CMIP6 models relative to CMIP5. However, unless the changes in model physics also reduce the uncertainties in the water vapor and albedo feedbacks, this study would suggest it is unlikely that the cloud feedback becomes a more important contributor to the WPS than the water vapor and albedo feedbacks. A similar statement is now added in the Discussion section, with the references provided by the reviewer added as well.

Some minor edits to consider:

- The abstract could be improved. Acronyms are not defined and there is not enough context of the study to be useful if someone has not read the paper.
- Line 37: define RCP
- Line 38: better expressed “..likely warm between 2.6 K and 4.8 K by the end of the 21st century relative to the end of the of the 20th century.”
- Line 71: define CMIP
- Line 83: with reference to Figure 1c and the statement that the sum of the partials roughly equals the total – there are some significant differences such as over the Southern Ocean that are physically significant and are worthy of study and improvement – might be worth simply recognizing this in the text – it is a notable feature throughout your analysis (e.g. Fig S6.)
- Line 99: and other section titles (L 119)/places replace “Dominate” and “Dominant”
- Line 120: specify “individual feedback processes” rather than individual processes
- Line 134: “water vapor feedbacks” rather than water feedbacks
- Line 343: notations are different in text and equation (ϕ)
- Table S1: also inconsistent notation between table caption and heading for Δ notations
- Table S3: What does the vertical resolution mean? Define P and S for non-modelers.

Response: We have made the suggested corrections accordingly. We pointed out the noticeable errors of the CFRAM analysis over the Southern Ocean.

Zonal-mean PRP at TOA ($W \cdot m^{-2}$)

Figure A1. Partial Radiative Energy Flux Perturbations at the TOA. Zonal means of partial radiative energy flux perturbations (PRP, W/m^2) given by all 26 models. Panels (a)–(f) are PRP associated with, respectively, the external forcing (EXT), Planck feedback (PL), surface albedo feedback (AL), water vapor feedback (WV), cloud feedback (CLD), and lapse rate feedback (LR). The solid line in each panel is the MME mean and the dashed thick line corresponds to the FGOALS-s2 model’s results.

Figure A2. Partial Surface Temperature Changes. Zonal means of CFRAM-derived partial surface temperature changes given by all 25 models. Panels (a)–(h) are CFRAM-decomposed partial surface temperature changes due to changes in, respectively, the external forcing (EXT), surface albedo (AL), water vapor (WV), clouds (CLD), atmospheric dynamics (ATM), ocean dynamics and heat storage (OCN), surface heat fluxes (HF), and ozone (O3). Panel (i) is the sum of (a)–(h) and (j) is the difference between Fig. 1b and the panel (i), corresponding to the error of the CFRAM analysis. The solid line in each panel is the MME mean and the dashed thick line corresponds to the FGOALS-s2 model’s results.

Peer Review File - Reviewers' comments, first round:

Reviewer #1 (Remarks to the Author):

This paper makes a nice case that the coupling among feedback processes in the climate system is important to better understand and represent accurately in models to decrease uncertainty in warming projections. The study uses statistical analyses (EOFs) of CMIP5 results to show that particular feedbacks are strongly coupled across the range of models. The result differ from previous studies owing to their use of geographical variations in the warming projection spread and feedback strengths as well as their use of a warming projection at the surface which provides insights that are not evident from global mean analyses done for top of the atmosphere changes in radiative fluxes.

Ultimately, the results show that a strong relationship between the ice-albedo feedback and water vapor feedback in the models leads to global patterns that are dependent on the strength of these two feedbacks and their relationship to each other. These feedbacks are also negatively related to the surface heat fluxes. The spread in the warming among the models is largely controlled by the strength of this group of feedbacks. While the absolute spread in the cloud feedback is large and has been identified as a major contributor to the spread in global mean warming projects, it is not systematically related to other processes in the model so may damp or enhance other feedbacks in space and time and thus is not the dominant contributor to the warming projection spread as it has been analyzed here.

The analysis is sound and well presented with all contributing data and methods clearly presented. The figures are clear. The analysis is complex so there are many figures but I think the authors have made the appropriate decisions about what to put in the main section of the paper versus the supplementary material.

I would recommend that this paper be published with attention to the editorial revisions listed below outside of one overarching concern.

In the discussion section the results are presented such that the largest contributors to WPS and largest uncertainties in climate projects are the same thing. Stated another way, could the uncertainty in the coupling of the cloud feedback with the other feedbacks change the fundamental relationship of this analysis enough that the spread in the cloud feedback is then the primary contributor to uncertainty? The message that this study presents – that reducing spread in model representations of ice-albedo feedback, water vapor feedback, and the relationship between the two will necessarily reduce model warming projecting uncertainty – is important. But the fact remains that there is large spread in the cloud feedback representations AND their relationships to other feedbacks and as currently represented there are dampening factors that reduce their contribution to the WPS analyzed in this study. A recent number of papers analyzing early outcomes from CMIP6 model runs (e.g., Gettelman et al 2019 GRL, Wyser et al 2019 GMD, Zelinka et al 2020 GRL) indicate that changes in model physics that affect cloud feedbacks are the largest contributors to increases in equilibrium climate sensitivity and thus the primary factor in changing projections of warming. The impact on ECS and WPS may be two different ways to assess warming projects that provide different estimates on uncertainty. I would like to hear the author's defense of their discussions and conclusions.

Some minor edits to consider:

- The abstract could be improved. Acronyms are not defined and there is not enough context of the study to be useful if someone has not read the paper.
- Line 37: define RCP
- Line 38: better expressed “..likely warm between 2.6 K and 4.8 K by the end of the 21st century relative to the end of the of the 20th century.”
- Line 71: define CMIP
- Line 83: with reference to Figure 1c and the statement that the sum of the partials roughly equals the total – there are some significant differences such as over the Southern Ocean that are physically significant and are worthy of study and improvement – might be worth simply recognizing this in the text – it is a notable feature throughout your analysis (e.g. Fig S6.)

- Line 99: and other section titles (L 119)/places replace "Dominate" and "Dominant"
- Line 120: specify "individual feedback processes" rather than individual processes
- Line 134: "water vapor feedbacks" rather than water feedbacks
- Line 343: notations are different in text and equation (ϕ)
- Table S1: also inconsistent notation between table caption and heading for $\langle \rangle$ notations
- Table S3: What does the vertical resolution mean? Define P and S for non-modelers.

Reviewer #2 (Remarks to the Author):

Hu et al. analyse model output of the former phase of the Coupled Model Intercomparison Project (CMIP5) to assess the pattern of warming in 2100 in the RCP8.5 scenario in relationship to feedbacks quantified at the surface. This study differs from previous analyses in two aspects: (1) feedbacks are defined at the surface, and include thus additional processes, and (2) the authors focus on the geographical distribution of warming, rather than on its global amount. The authors analyse CMIP5 models that have been studied intensely before, the results in the literature assessing top-of-atmosphere, global quantities yield very consistent results that are adequately reported in the introduction: importance of the cloud feedback for the global temperature uncertainty, importance of surface albedo and lapse-rate feedback for Arctic amplification. In title and discussions the present manuscript seems to suggest that the previous literature is called into question by the results presented here. However, little evidence is provided. Rather, it seems very plausible that the difference in conclusion is due to the difference in question asked. One is the surface perspective: this is a very different approach compared to the top of the atmosphere (TOA). It is the energy budget at TOA that determines climate. Some feedbacks act similarly at surface and TOA, most show distinct differences in their effects. The decomposition is much less straightforward at the surface, where also the fluxes from below (land or ocean surface) need to be taken into account. This makes the interpretation more difficult. The other aspect is the focus on pattern rather than overall magnitude. The cloud feedback has a noisy and not very distinct spatial pattern. The surface albedo effect has a very distinct pattern (strong and positive in high latitudes, zero in low latitudes) as has the water vapour feedback (exponentially related to surface temperature). It is thus not surprising that these two correlate well with distinct warming patterns.

Overall, the study is well conducted and instructive. I question, however, the way the results are presented, which seemingly construct discrepancies with previous literature due to the aim to publish in a non-specialized journal. So I would urge the authors to re-formulate their findings in a more balanced way, in which case the study may very well be worth publication, but better so in a specialized journal.

Reviewer #3 (Remarks to the Author):

Review of " A Less Cloudy Picture of the Inter-Model Spread in Future Global Warming Projections " by Hu et al. [Manuscript Number: NCOMMS-19-42302].

This paper investigates the spread in surface temperature response from CMIP5 models under RCP 8.5, and uses the "CFRAM" methodology to isolate surface temperature change contributions from individual processes including water vapour, clouds, turbulent fluxes, oceanic convergence, surface albedo et cetera. The paper uses an EOF approach to define the major meridional patterns of temperature change, then correlates these with the spreads in the surface temperature change components. It concludes that water vapour is the dominant feedback causing change at low latitudes, with surface albedo dominant at high. It concludes that reductions in uncertainties in these two feedbacks should be a priority for reducing spread in global temperature projections.

The authors are to be commended for exploring the surface contributions to temperature change, and relating these to individual processes and feedbacks, with the aim of understanding the causes of the meridional pattern changes. The CFRAM methodology is suited to this, and the analysis for

the most part is a sound application of this. I do have some significant issues, however with the conclusions reached, particularly with regard to the role and importance of the individual feedbacks, and the consistency or otherwise with findings elsewhere that take a top of atmosphere view. There are also a range of clarifications needed. These issues would need to be addressed before the paper was suitable for publication.

Major comments:

(1). A much clearer discussion of the contrast between surface and TOA feedbacks needs to be made. An extremely large number of papers and studies, along with the last three IPCC reports, conclude that cloud uncertainties dominate the uncertainty in the climate sensitivity and the WPS at a global scale. Furthermore, the processes controlling (top of atmosphere) water vapour/lapse rate feedbacks are relatively well understood, as is the anti-correlation between them, and the large uncertainty in cloud processes and their impact on the Earth's atmospheric radiative balance (both rapid response and feedback) are clearly established. The conclusions of this paper sit at variance with this – or at least are expressed to be at variance with this, therefore the authors must establish clearly how their surface energy budget perspective provides a different interpretation. I suspect there is ultimately consistency between the two, because they are addressing different issues – top of atmosphere response of processes controlling global temperature response, and net energy imbalances controlling surface meridional warming patterns. The authors need to clearly articulate that these are different and what drives the differences.

One place to start could be to also perform a TOA analysis along the same lines and compare the results. My understanding is that CFRAM can produce such an analysis, and therefore the contrast could be made. Another issue to address is, since the lapse rate feedback is not included in CFRAM as a separate process, is this a major factor helping to drive the differences between surface/TOA? The authors also need to discuss these differences at length, so that the differences in TOA/surface interpretations can be understood.

(2). The above comment related to global differences in temperature response. At the meridional level there is also inconsistency between the current results and those of earlier studies (e.g. Vial et al, Bonan et al., 2018). For example the Bonan et al states that "cloud feedbacks, in particular, contribute the greatest source of warming uncertainty in most regions". The present paper needs to address/explain its differing conclusions at the meridional and at the global level.

(3) The authors should clarify the physical meaning of surface non-radiative "feedbacks". For example, how do changes in convective or advective convergence "feed back" on the surface temperature change? For example does increased surface warming/cooling due to changes in turbulent fluxes in turn reinforce/oppose those flux changes and further increase surface warming (i.e. feed back' on it?? At the TOA the process is clear, but at the surface less so.

(4) Earlier studies have found that because of nonlinearities in radiative response to clouds (overlap etc), using long-term average for clouds do not provide an accurate representation of the radiative impact of cloud changes (e.g. early papers on the Partial Radiative Perturbation approach – Wetherald and Manabe, 1988, and a number of later papers). To overcome this, PRP typically uses daily cloud fractions in radiation calculations. Yet the approach here uses 50 year mean clouds in its calculations (e.g. lines 279+). Considering critical conclusions are being made about the strength and spread of the cloud feedback, the authors need to establish that cloud feedbacks are accurately represented from CFRAM using long-term means.

(5). The authors need to clarify that it is actually spread in water vapour "feedback" that causes the spread in surface temperature change. I mean here the radiative response from water vapour changes per degree of surface warming, not just the radiative changes resulting from water vapour changes. A surface temperature increase (particularly in the tropics), will always be associated with warmer, moister air at low levels which will more strongly radiate to the surface. So a warmer surface will be necessarily associated with stronger down welling longwave radiation from water vapour, but it doesn't mean that water vapour is causing that extra warming (it may simply be a result of the warming). To paraphrase: is the water vapour term normalised by the surface temperature change, or is it just the absolute change in down welling radiation from water vapour changes?

(6). The paper is pretty vague about "positive coupling" between e.g. water vapour and surface albedo feedbacks. Please define precisely what this means, and what the evidence is for it.

(7) The paper discusses the range in the global WPS, and the meridional pattern of WPS. To aid

the reader, the authors should make it clearer which they are talking about and when. Jumping between the two is confusing. E.g. in lines 104 to 118. I suggest the paper has different sections on global mean WPS spread and the spread in the meridional patterns (i.e. regional WPS).

(8) in summary a number of significant issues need to be addressed before the claim made in the last sentence of the abstract, or in lines 165-167 can be substantiated. I encourage the authors to address these issues, as their approach is an interesting and worthwhile one and a surface perspective on feedbacks and feedback spread would be extremely useful in the literature.

(9) I found a number of figures hard to digest. These include figure 4 and supplementary figures 3, 4 and 5. It would help the reader if these models were not defined in numerical order (i.e. alphabetical order) but something more meaningful to the discussion -- strongest to weakest temperature response, or some other relevant physical parameter.

Minor comments and corrections

(1) abstract, line 24: define "WPS"

(2) abstract line 24: the authors need to be much more precise here: it is change to surface temperature, and it is the meridional pattern, not the global value that is being discussed (compare with figure S8, where the explained variance of the surface pattern is much lower).

(3) Abstract line 26: see major comments above. The abstract talks of these feedbacks "driving uncertainties" where there needs to be much more discussion of causality

(4) abstract line 32: missing the word "only" before "... A little"

(5) line 49: need to insert the word "meridional" before "warming projection"

(6) line 74: delete 'an' before easy.

(7) line 99: dominate -> dominant

(8) line 119: isolations -> isolation

(9) line 123: insert 'regional' before WPS to contrast it with the global discussion

(10) line 125-128: please clarify how it is established that the spatial pattern or warming spread is caused by a "strong positive coupling" between these feedbacks

(11) define PWA in line 130

(12) line 138: as discussed above, why is the energy convergence change associated with surface turbulent heat fluxes a "feedback". Isn't it just a component of the surface energy budget?

(13) line 160: discussions -> discussion

(14) lines 165-7: see major comments above. Surely this should have the words 'regional patterns of' in front of 'climate change projections'

(15) line 406: scattering -> scatter

(16) supplementary table 3: what does "P" mean under the vertical resolution column? (Also what does S stand for? Presumably surface?)

(17) Please clarify more carefully the purpose of supplementary figures 3 and 4.

(18) I found the caption and figure S5 and figure S6 hard to understand.